# ROS: A GNN-based Relax-Optimize-and-Sample Framework for Max-$k$-Cut Problems

**Yeqing Qiu** [1 2]   **Ye Xue** [1 2]   **Akang Wang** [1 2]   **Yiheng Wang** [1 2]   **Qingjiang Shi** [1 3]   **Zhi-Quan Luo** [1 2]

## Abstract

The Max-$k$-Cut problem is a fundamental combinatorial optimization challenge that generalizes the classic $\mathcal{NP}$-complete Max-Cut problem. While relaxation techniques are commonly employed to tackle Max-$k$-Cut, they often lack guarantees of equivalence between the solutions of the original problem and its relaxation. To address this issue, we introduce the Relax-Optimize-and-Sample (ROS) framework. In particular, we begin by relaxing the discrete constraints to the continuous probability simplex form. Next, we pre-train and fine-tune a graph neural network model to efficiently optimize the relaxed problem. Subsequently, we propose a sampling-based construction algorithm to map the continuous solution back to a high-quality Max-$k$-Cut solution. By integrating geometric landscape analysis with statistical theory, we establish the consistency of function values between the continuous solution and its mapped counterpart. Extensive experimental results on random regular graphs, the Gset benchmark, and the real-world datasets demonstrate that the proposed ROS framework effectively scales to large instances with up to $20,000$ nodes in just a few seconds, outperforming state-of-the-art algorithms. Furthermore, ROS exhibits strong generalization capabilities across both in-distribution and out-of-distribution instances, underscoring its effectiveness for large-scale optimization tasks.

## 1. Introduction

The *Max-$k$-Cut problem* involves partitioning the vertices of a graph into $k$ disjoint subsets in such a way that the total weight of edges between vertices in different subsets is maximized. This problem represents a significant challenge in combinatorial optimization and finds applications across various fields, including telecommunication networks (Eisenblätter, 2002; Gui et al., 2019), data clustering (Poland & Zeugmann, 2006; Ly et al., 2023), and theoretical physics (Cook et al., 2019; Coja-Oghlan et al., 2022). The Max-$k$-Cut problem is known to be $\mathcal{NP}$-complete, as it generalizes the well-known *Max-Cut problem*, which is one of the 21 classic $\mathcal{NP}$-complete problems identified by Karp (1972).

Significant efforts have been made to develop methods for solving Max-$k$-Cut problems (Nath & Kuhnle, 2024). Ghaddar et al. (2011) introduced an exact branch-and-cut algorithm based on semi-definite programming, capable of handling graphs with up to 100 vertices. For larger instances, various polynomial-time approximation algorithms have been proposed. Goemans & Williamson (1995) addressed the Max-Cut problem by first solving a semi-definite relaxation to obtain a fractional solution, then applying a randomization technique to convert it into a feasible solution, resulting in a $0.878$-approximation algorithm. Building on this, Frieze & Jerrum (1997) extended the approach to Max-$k$-Cut, offering feasible solutions with approximation guarantees. de Klerk et al. (2004) further improved these guarantees, while Shinde et al. (2021) optimized memory usage. Despite their strong theoretical performance, these approximation algorithms involve solving computationally intensive semi-definite programs, rendering them impractical for large-scale Max-$k$-Cut problems. A variety of heuristic methods have been developed to tackle the scalability challenge. For the Max-Cut problem, Burer et al. (2002) proposed rank-two relaxation-based heuristics, and Goudet et al. (2024) introduced a meta-heuristic approach using evolutionary algorithms. For Max-$k$-Cut, heuristics such as genetic algorithms (Li & Wang, 2016), greedy search (Gui et al., 2019), multiple operator heuristics (Ma & Hao, 2017), and local search (Garvardt et al., 2023) have been proposed. While these heuristics can handle much larger Max-$k$-Cut instances, they often struggle to balance efficiency and solution quality.

Recently, *machine learning* techniques have gained attention for enhancing optimization algorithms (Bengio et al.,

[1]Shenzhen Research Institute of Big Data, Shenzhen, China. [2]The Chinese University of Hong Kong, Shenzhen, China. [3]Tongji University, Shanghai, China. Correspondence to: Ye Xue <xueye@cuhk.edu.cn>.

*Proceedings of the $42^{nd}$ International Conference on Machine Learning*, Vancouver, Canada. PMLR 267, 2025. Copyright 2025 by the author(s).

2021; Gasse et al., 2022; Chen et al., 2024). Several studies, including Khalil et al. (2017); Barrett et al. (2020); Chen et al. (2020); Barrett et al. (2022), framed the Max-Cut problem as a sequential decision-making process, using reinforcement learning to train policy networks for generating feasible solutions. However, RL-based methods often suffer from extensive sampling efforts and increased complexity in action space when extended to Max-$k$-Cut, and hence entails significantly longer training and testing time. Karalias & Loukas (2020) focuses on subset selection, including Max-Cut as a special case. It trains a *graph neural network* (GNN) to produce a distribution over subsets of nodes of an input graph by minimizing a probabilistic penalty loss function. After the network has been trained, a randomized algorithm is employed to sequentially decode a valid Max-Cut solution from the learned distribution. A notable advancement by Schuetz et al. (2022) reformulated Max-Cut as a quadratic unconstrained binary optimization (QUBO), removing binarity constraints to create a differentiable loss function. This loss function was used to train a GNN, followed by a simple projection onto integer variables after unsupervised training. The key feature of this approach is solving the Max-Cut problem during the training phase, eliminating the need for a separate testing stage. Although this method can produce high-quality solutions for Max-Cut instances with millions of nodes, the computational time remains significant due to the need to optimize a parameterized GNN from scratch. The work of Tönshoff et al. (2023) first formulated the Max-Cut problem as a *Constraint Satisfaction Problem* (CSP) and then proposed a novel GNN-based reinforcement learning approach. This method outperforms prior neural combinatorial optimization techniques and conventional search heuristics. However, to the best of our knowledge, it is limited to unweighted Max-$k$-Cut problems. NeuroCUT (Shah et al., 2024) is a partitioning method based on reinforcement learning, whereas DGCLUSTER (Bhowmick et al., 2024) and DMoN (Tsitsulin et al., 2023) utilize GNNs to optimize clustering objectives. However, these methods are specifically designed for graph clustering, which focuses on minimizing inter-cluster connections—contrary to Max-$k$-Cut, where the goal is to maximize inter-partition connections. Consequently, they are not directly applicable to our problem. Although NeuroCUT claims support for arbitrary objective functions, its node selection heuristics are tailored exclusively for graph clustering, rendering it unsuitable for Max-$k$-Cut.

In this work, we propose a GNN-based *Relax-Optimize-and-Sample* (ROS) framework for efficiently solving the Max-$k$-Cut problem with arbitrary edge weights. The framework is depicted in Figure 1. Initially, the Max-$k$-Cut problem is formulated as a discrete optimization task. To handle this, we introduce *probability simplex relaxations*, transforming the discrete problem into a continuous one. We then op-

timize the relaxed formulation by training parameterized GNNs in an unsupervised manner. To further improve efficiency, we apply *transfer learning*, utilizing pre-trained GNNs to warm-start the training process. Finally, we refine the continuous solution using a *random sampling algorithm*, resulting in high-quality Max-$k$-Cut solutions.

The key contributions of our work are summarized as follows:

- **Novel Framework.** We propose a scalable ROS framework tailored to the weighted Max-$k$-Cut problem with arbitrary signs, built on solving continuous relaxations using efficient learning-based techniques.

- **Theoretical Foundations.** We conduct a rigorous theoretical analysis of both the relaxation and sampling steps. By integrating geometric landscape analysis with statistical theory, we demonstrate the consistency of function values between the continuous solution and its sampled discrete counterpart.

- **Superior Performance.** Comprehensive experiments on public benchmark datasets show that our framework produces high-quality solutions for Max-$k$-Cut instances with up to $20,000$ nodes in just a few seconds. Our approach significantly outperforms state-of-the-art algorithms, while also demonstrating strong generalization across various instance types.

## 2. Preliminaries

### 2.1. Max-$k$-Cut Problems

Let $\mathcal{G} = (\mathcal{V}, \mathcal{E})$ represent an undirected graph with vertex set $\mathcal{V}$ and edge set $\mathcal{E}$. Each edge $(i, j) \in \mathcal{E}$ is assigned an arbitrary weight $\boldsymbol{W}_{ij} \in \mathbb{R}$, which can have any sign. A *cut* in $\mathcal{G}$ refers to a partition of its vertex set. The Max-$k$-Cut problem involves finding a $k$-partition $(\mathcal{V}_1, \ldots, \mathcal{V}_k)$ of the vertex set $\mathcal{V}$ such that the sum of the weights of the edges between different partitions is maximized.

To represent this partitioning, we employ a $k$-dimensional one-hot encoding scheme. Specifically, we define a $k \times N$ matrix $\boldsymbol{X} \in \mathbb{R}^{k \times N}$ where each column represents a one-hot vector. The Max-$k$-Cut problem can be formulated as:

$$\max_{\boldsymbol{X} \in \mathbb{R}^{k \times N}} \quad \frac{1}{2} \sum_{i=1}^{N} \sum_{j=1}^{N} \boldsymbol{W}_{ij} \left(1 - \boldsymbol{X}_{\cdot i}^{\top} \boldsymbol{X}_{\cdot j}\right) \tag{1}$$
$$\text{s. t.} \quad \boldsymbol{X}_{\cdot j} \in \{\boldsymbol{e}_1, \boldsymbol{e}_2, \ldots, \boldsymbol{e}_k\} \qquad \forall j \in \mathcal{V},$$

where $\boldsymbol{X}_{\cdot j}$ denotes the $j^{th}$ column of $\boldsymbol{X}$, $\boldsymbol{W}$ is a symmetric matrix with zero diagonal entries, and $\boldsymbol{e}_\ell \in \mathbb{R}^k$ is a one-hot vector with the $\ell^{th}$ entry set to 1. This formulation aims to maximize the total weight of edges between different partitions, ensuring that each node is assigned to exactly one

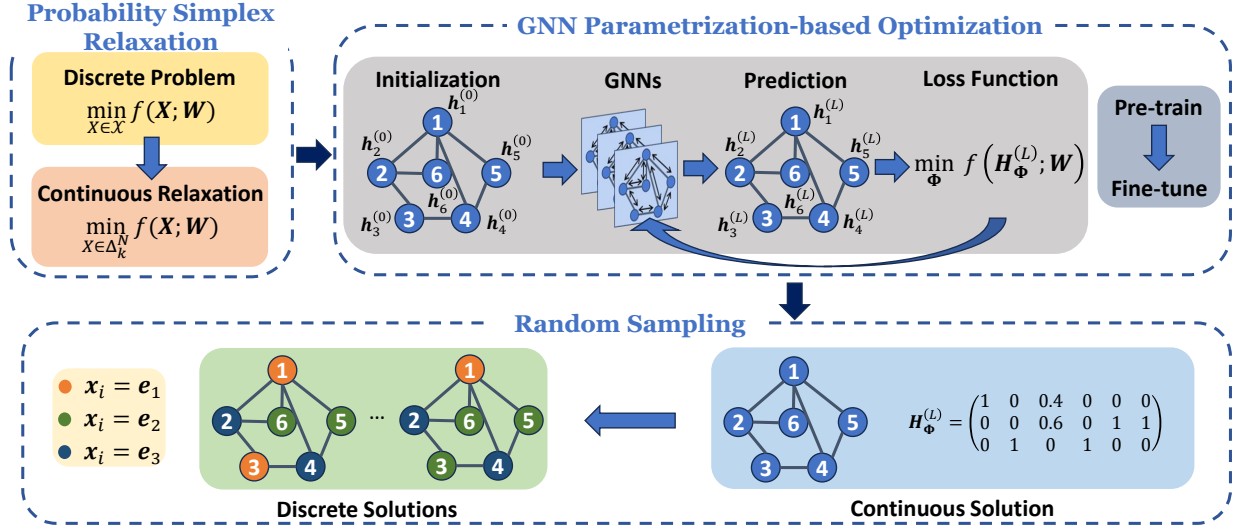

Figure 1: The Relax-Optimize-and-Sample framework.

partition, represented by the one-hot encoded vectors. We remark that weighted Max-$k$-Cut problems with arbitrary signs is a generalization of classic Max-Cut problems and arise in many interesting applications (De Simone et al., 1995; Poland & Zeugmann, 2006; Hojny et al., 2021).

## 2.2. Graph Neural Networks

GNNs are powerful tools for learning representations from graph-structured data. GNNs operate by iteratively aggregating information from a node's neighbors, enabling each node to capture increasingly larger sub-graph structures as more layers are stacked. This process allows GNNs to learn complex patterns and relationships between nodes, based on their local connectivity.

At the initial layer ($l = 0$), each node $i \in \mathcal{V}$ is assigned a feature vector $\boldsymbol{h}_i^{(0)}$, which typically originates from node features or labels. The representation of node $i$ is then recursively updated at each subsequent layer through a parametric aggregation function $f_{\boldsymbol{\Phi}^{(l)}}$, defined as:

$$\boldsymbol{h}_i^{(l)} = f_{\boldsymbol{\Phi}^{(l)}}\left(\boldsymbol{h}_i^{(l-1)}, \{\boldsymbol{h}_j^{(l-1)} : j \in \mathcal{N}(i)\}\right), \quad (2)$$

where $\boldsymbol{\Phi}^{(l)}$ represents the trainable parameters at layer $l$, $\mathcal{N}(i)$ denotes the set of neighbors of node $i$, and $\boldsymbol{h}_i^{(l)}$ is the node's embedding at layer $l$ for $l \in \{1, 2, \cdots, L\}$. This iterative process enables the GNN to propagate information throughout the graph, capturing both local and global structural properties.

## 3. A Relax-Optimize-and-Sample Framework

In this work, we leverage continuous optimization techniques to tackle Max-$k$-Cut problems, introducing a novel ROS framework. Acknowledging the inherent challenges of discrete optimization, we begin by relaxing the problem to probability simplices and concentrate on optimizing this relaxed version. To achieve this, we propose a machine learning-based approach. Specifically, we model the relaxed problem using GNNs, pre-training the GNN on a curated graph dataset before fine-tuning it on the specific target instance. After obtaining high-quality solutions to the relaxed continuous problem, we employ a random sampling procedure to derive a discrete solution that preserves the same objective value.

### 3.1. Probability Simplex Relaxations

To simplify the formulation of the problem (1), we remove constant terms and negate the objective function, yielding an equivalent formulation expressed as follows:

$$\min_{\boldsymbol{X} \in \mathcal{X}} \quad f(\boldsymbol{X}; \boldsymbol{W}) \coloneqq \mathrm{Tr}(\boldsymbol{X}\boldsymbol{W}\boldsymbol{X}^{\top}), \quad (\mathbf{P})$$

where $\mathcal{X} \coloneqq \left\{\boldsymbol{X} \in \mathbb{R}^{k \times N} : \boldsymbol{X}_{\cdot j} \in \{\boldsymbol{e}_1, \boldsymbol{e}_2, \ldots, \boldsymbol{e}_k\}, \forall j \in \mathcal{V}\right\}$. It is important to note that the matrix $\boldsymbol{W}$ is indefinite due to its diagonal entries being set to zero.

Given the challenges associated with solving the discrete problem $\mathbf{P}$, we adopt a naive relaxation approach, obtaining the convex hull of $\mathcal{X}$ as the Cartesian product of $N$ $k$-dimensional probability simplices, denoted by $\Delta_k^N$. Consequently, the discrete problem $\mathbf{P}$ is relaxed into the follow-

ing continuous optimization form:

$$\min_{\boldsymbol{X} \in \Delta_k^N} \quad f(\boldsymbol{X}; \boldsymbol{W}). \tag{$\overline{\mathbf{P}}$}$$

Before optimizing problem $\overline{\mathbf{P}}$, we will characterize its *geometric landscape*. To facilitate this, we introduce the following definition.

**Definition 3.1.** Let $\overline{\boldsymbol{X}}$ denote a point in $\Delta_k^N$. We define the neighborhood induced by $\overline{\boldsymbol{X}}$ as follows:

$$\mathcal{N}(\overline{\boldsymbol{X}}) := \left\{ \boldsymbol{X} \in \Delta_k^N \; \middle| \; \sum_{i \in \mathcal{K}(\overline{\boldsymbol{X}}_{\cdot j})} \boldsymbol{X}_{ij} = 1, \quad \forall j \in \mathcal{V} \right\},$$

where $\mathcal{K}(\overline{\boldsymbol{X}}_{\cdot j}) := \{i \in \{1, \dots, k\} \mid \overline{\boldsymbol{X}}_{ij} > 0\}$.

The set $\mathcal{N}(\overline{\boldsymbol{X}})$ represents a neighborhood around $\overline{\boldsymbol{X}}$, where each point in $\mathcal{N}(\overline{\boldsymbol{X}})$ can be derived by allowing each non-zero entry of the matrix $\overline{\boldsymbol{X}}$ to vary freely, while the other entries are set to zero. Utilizing this definition, we can establish the following theorem.

**Theorem 3.2.** *Let $\boldsymbol{X}^\star$ denote a globally optimal solution to $\overline{\mathbf{P}}$, and let $\mathcal{N}(\boldsymbol{X}^\star)$ be its induced neighborhood. Then*

$$f(\boldsymbol{X}; \boldsymbol{W}) = f(\boldsymbol{X}^\star; \boldsymbol{W}), \quad \forall \boldsymbol{X} \in \mathcal{N}(\boldsymbol{X}^\star).$$

Theorem 3.2 states that for a globally optimal solution $\boldsymbol{X}^\star$, every point within its neighborhood $\mathcal{N}(\boldsymbol{X}^\star)$ shares the same objective value as $\boldsymbol{X}^\star$, thus forming a *basin* in the geometric landscape of $f(\boldsymbol{X}; \boldsymbol{W})$. If $\boldsymbol{X}^\star \in \mathcal{X}$ (i.e., an integer solution), then $\mathcal{N}(\boldsymbol{X}^\star)$ reduces to the singleton set $\{\boldsymbol{X}^\star\}$. Conversely, if $\boldsymbol{X}^\star \notin \mathcal{X}$, there exist $\prod_{j \in \mathcal{V}} |\mathcal{K}(\boldsymbol{X}_{\cdot j}^\star)|$ unique integer solutions within $\mathcal{N}(\boldsymbol{X}^\star)$ that maintain the same objective value as $\boldsymbol{X}^\star$. This indicates that once a globally optimal solution to the relaxed problem $\overline{\mathbf{P}}$ is identified, it becomes straightforward to construct an optimal solution for the original problem $\mathbf{P}$ that preserves the same objective value.

According to Carlson & Nemhauser (1966), among all globally optimal solutions to the relaxed problem $\overline{\mathbf{P}}$, the integer solution always exists. Theorem 3.2 extends this result, indicating that if the globally optimal solution is fractional, we can provide a straightforward method to derive its integer counterpart. We remark that it is highly non-trivial to guarantee that the feasible Max-$k$-Cut solution obtained from the relaxation one has the same quality.

**Example.** Consider a Max-Cut problem ($k = 2$) associated with the weight matrix $\boldsymbol{W}$. We optimize its relaxation and obtain the optimal solution $\boldsymbol{X}^\star$.

$$\boldsymbol{W} := \begin{pmatrix} 0 & 1 & 1 \\ 1 & 0 & 1 \\ 1 & 1 & 0 \end{pmatrix}, \boldsymbol{X}^\star := \begin{pmatrix} p & 1 & 0 \\ 1-p & 0 & 1 \end{pmatrix},$$

where $p \in [0, 1]$. From the neighborhood $\mathcal{N}(\boldsymbol{X}^\star)$, we can identify the following integer solutions that maintain the same objective value.

$$\boldsymbol{X}_1^\star = \begin{pmatrix} 0 & 1 & 0 \\ 1 & 0 & 1 \end{pmatrix}, \boldsymbol{X}_2^\star = \begin{pmatrix} 1 & 1 & 0 \\ 0 & 0 & 1 \end{pmatrix}.$$

Given that $\overline{\mathbf{P}}$ is a non-convex program, identifying its global minimum is challenging. Consequently, the following two critical questions arise.

**Q1.** Since solving $\overline{\mathbf{P}}$ to global optimality is $\mathcal{NP}$-hard, how to efficiently optimize $\overline{\mathbf{P}}$ for high-quality solutions?

**Q2.** Given $\overline{\boldsymbol{X}} \in \Delta_k^N \setminus \mathcal{X}$ as a high-quality solution to $\overline{\mathbf{P}}$, can we construct a feasible solution $\hat{\boldsymbol{X}} \in \mathcal{X}$ to $\mathbf{P}$ such that $f(\hat{\boldsymbol{X}}; \boldsymbol{W}) = f(\overline{\boldsymbol{X}}; \boldsymbol{W})$?

We provide a positive answer to **Q2** in Section 3.2, while our approach to addressing **Q1** is deferred to Section 3.3.

### 3.2. Random Sampling

Let $\overline{\boldsymbol{X}} \in \Delta_k^N \setminus \mathcal{X}$ be a feasible solution to the relaxation $\overline{\mathbf{P}}$. Our goal is to construct a feasible solution $\boldsymbol{X} \in \mathcal{X}$ for the original problem $\mathbf{P}$, ensuring that the corresponding objective values are equal. Inspired by Theorem 3.2, we propose a *random sampling* procedure, outlined in Algorithm 1. In this approach, we sample each column $\boldsymbol{X}_{\cdot i}$ of the matrix $\boldsymbol{X}$ from a categorical distribution characterized by the event probabilities $\overline{\boldsymbol{X}}_{\cdot i}$ (denoted as $\text{Cat}(\boldsymbol{x}; \boldsymbol{p} = \overline{\boldsymbol{X}}_{\cdot i})$ in Step 3 of Algorithm 1). This randomized approach yields a feasible solution $\hat{\boldsymbol{X}}$ for $\mathbf{P}$. However, since Algorithm 1 incorporates randomness in generating $\hat{\boldsymbol{X}}$ from $\overline{\boldsymbol{X}}$, the value of $f(\hat{\boldsymbol{X}}; \boldsymbol{W})$ becomes random as well. This raises the critical question: is this value greater or lesser than $f(\overline{\boldsymbol{X}}; \boldsymbol{W})$? We address this question in Theorem 3.3.

---

**Algorithm 1** Random Sampling

---

1: **Input:** $\overline{\boldsymbol{X}} \in \Delta_k^N$
2: **for** $i = 1$ to $N$ **do**
3: $\quad \hat{\boldsymbol{X}}_{\cdot i} \sim \text{Cat}(\boldsymbol{x}; \boldsymbol{p} = \overline{\boldsymbol{X}}_{\cdot i})$
4: **end for**
5: **Output:** $\hat{\boldsymbol{X}} \in \mathcal{X}$

---

**Theorem 3.3.** *Let $\overline{\boldsymbol{X}}$ and $\hat{\boldsymbol{X}}$ denote the input and output of Algorithm 1, respectively. Then, we have $\mathbb{E}_{\hat{\boldsymbol{X}}}[f(\hat{\boldsymbol{X}}; \boldsymbol{W})] = f(\overline{\boldsymbol{X}}; \boldsymbol{W})$.*

Theorem 3.3 states that $f(\hat{\boldsymbol{X}}; \boldsymbol{W})$ is equal to $f(\overline{\boldsymbol{X}}; \boldsymbol{W})$ in expectation. This implies that the random sampling procedure operates on a fractional solution, yielding Max-$k$-Cut feasible solutions with the same objective values in a probabilistic sense. While the Lovász-extension-based

method (Bach, 2013) also offers a framework for continuous relaxation, achieving similar theoretical results for arbitrary $k$ and edge weights $\boldsymbol{W}_{i,j} \in \mathbb{R}$ is not always guaranteed. In practice, we execute Algorithm 1 $T$ times and select the solution with the lowest objective value of $f$ as our best result. We remark that the theoretical interpretation in Theorem 3.3 distinguishes our sampling algorithm from the existing ones in the literature (Karalias & Loukas, 2020; Tönshoff et al., 2021; Michael et al., 2024).

### 3.3. GNN Parametrization-Based Optimization

To solve the problem $\overline{\mathbf{P}}$, we propose an efficient learning-to-optimize (L2O) method based on GNN parametrization. This approach reduces the laborious iterations typically required by classical optimization methods (e.g., mirror descent). Additionally, we introduce a "pre-train + fine-tune" strategy, where the model is endowed with prior graph knowledge during the pre-training phase, significantly decreasing the computational time required to optimize $\overline{\mathbf{P}}$.

**GNN Parametrization.** The Max-$k$-Cut problem can be framed as a node classification task, allowing us to leverage GNNs to aggregate node features, and obtain high-quality solutions. Initially, we assign a random embedding $\boldsymbol{h}_i^{(0)}$ to each node $i$ in the graph $\mathcal{G}$. We adopt the GNN architecture proposed by Morris et al. (2019), utilizing an $L$-layer GNN with updates at layer $l$ given by:

$$\boldsymbol{h}_i^{(l)} := \sigma \left( \boldsymbol{\Phi}_1^{(l)} \boldsymbol{h}_i^{(l-1)} + \boldsymbol{\Phi}_2^{(l)} \sum_{j \in \mathcal{N}(i)} \boldsymbol{W}_{ji} \boldsymbol{h}_j^{(l-1)} \right),$$

where $\sigma(\cdot)$ is an activation function, and $\boldsymbol{\Phi}_1^{(l)}$ and $\boldsymbol{\Phi}_2^{(l)}$ are the trainable parameters at layer $l$. This formulation facilitates efficient learning of node representations by leveraging both node features and the underlying graph structure. After processing through $L$ layers of GNN, we obtain the final output $\boldsymbol{H}_{\boldsymbol{\Phi}}^{(L)} := [\boldsymbol{h}_1^{(L)}, \ldots, \boldsymbol{h}_N^{(L)}] \in \mathbb{R}^{k \times N}$. A softmax activation function is applied in the last layer to ensure $\boldsymbol{H}_{\boldsymbol{\Phi}}^{(L)} \in \Delta_k^N$, making the final output feasible for $\overline{\boldsymbol{P}}$.

**"Pre-train + Fine-tune" Optimization.** We propose a "pre-train + fine-tune" framework for learning the trainable weights of GNNs. Initially, the model is trained on a collection of pre-collected datasets to produce a pre-trained model. Subsequently, we fine-tune this pre-trained model for each specific testing instance. This approach equips the model with prior knowledge of graph structures during the pre-training phase, significantly reducing the overall solving time. Furthermore, it allows for out-of-distribution generalization due to the fine-tuning step.

In the pre-training phase, the trainable parameters $\boldsymbol{\Phi} := (\boldsymbol{\Phi}_1^{(1)}, \boldsymbol{\Phi}_2^{(1)}, \ldots, \boldsymbol{\Phi}_1^{(L)}, \boldsymbol{\Phi}_2^{(L)})$ are optimized using the Adam optimizer with *random initialization*, targeting the objective

$$\min_{\boldsymbol{\Phi}} \quad \mathcal{L}_{\text{pre-training}}(\boldsymbol{\Phi}) := \frac{1}{M} \sum_{m=1}^{M} f(\boldsymbol{H}_{\boldsymbol{\Phi}}^{(L)}; \boldsymbol{W}_{\text{train}}^{(m)}),$$

where $\mathcal{D} := \{\boldsymbol{W}_{\text{train}}^{(1)}, \ldots, \boldsymbol{W}_{\text{train}}^{(M)}\}$ represents the pre-training dataset. In the fine-tuning phase, for a problem instance $\boldsymbol{W}_{\text{test}}$, the Adam optimizer seeks to solve

$$\min_{\boldsymbol{\Phi}} \quad \mathcal{L}_{\text{fine-tuning}}(\boldsymbol{\Phi}) := f(\boldsymbol{H}_{\boldsymbol{\Phi}}^{(L)}; \boldsymbol{W}_{\text{test}}),$$

initialized with the pre-trained parameters.

Moreover, to enable the GNN model to fully adapt to specific problem instances, the pre-training phase can be omitted, enabling the model to be directly trained and tested on the same instance. While this direct approach may necessitate more computational time, it often results in improved performance regarding the objective function. Consequently, users can choose to include a pre-training phase based on the specific requirements of their application scenarios.

## 4. Experiments

### 4.1. Experimental Settings

We compare the performance of ROS against traditional methods as well as L2O algorithms for solving the Max-$k$-Cut problem. Additionally, we assess the impact of the "Pre-train" stage in the GNN parametrization-based optimization. The source code is available at `https://github.com/NetSysOpt/ROS`.

**Baseline Algorithms.** We denote our proposed algorithms by ROS and compare them against both traditional algorithms and L2O methods. When the pre-training step is skipped, we refer to our algorithm as ROS-vanilla. The following traditional Max-$k$-Cut algorithms are considered as baselines: (i) GW (Goemans & Williamson, 1995): an method with a 0.878-approximation guarantee based on semi-definite relaxation; (ii) BQP (Gui et al., 2019): a local search method designed for binary quadratic programs; (iii) Genetic (Li & Wang, 2016): a genetic algorithm specifically for Max-$k$-Cut problems; (iv) MD: a mirror descent algorithm that addresses the relaxed problem $\overline{\mathbf{P}}$ with a convergence tolerance at $10^{-8}$ and adopts the same random sampling procedure; (v) LPI (Goudet et al., 2024): an evolutionary algorithm featuring a large population organized across different islands; (vi) MOH (Ma & Hao, 2017): a heuristic algorithm based on multiple operator heuristics, employing various distinct search operators within the search phase. For the L2O method, we primarily examine the state-of-the-art baseline algorithms: (vii) PI-GNN (Schuetz et al., 2022): an unsupervised method for QUBO problems, which can model the

weighted Max-Cut problem, delivering commendable performance. (viii) `ECO-DQN` (Barrett et al., 2020): a reinforcement L2O method introducing test-time exploratory refinement for Max-Cut problems. (ix) `ANYCSP` (Tönshoff et al., 2023): an unsupervised GNN-based search heuristic for CSPs, which can model the unweighted Max-$k$-Cut problem, leveraging a compact graph representation and global search action with the default time limit of 180 seconds.

**Datasets.** We conduct experiments on the following datasets.

- $r$-**Random regular graphs** (Schuetz et al., 2022): Each node has the same degree $r$. Edge weights are either 0 or 1.

- **Gset** (Ye, 2003): A well-known Max-$k$-Cut benchmark comprising toroidal, planar, and random graphs with $800 \sim 20,000$ nodes and edge densities between 2% and 6%. Edge weights are either 0 or $\pm 1$.

- **COLOR** (Micheal, 2002): A collection of dense graphs derived from literary texts, where nodes represent characters and edges indicate co-occurrence. These graphs have large chromatic numbers ($\chi \approx 10$), making them suitable for Max-$k$-Cut. Edge weights are either 0 or 1.

- **Bitcoin-OTC** (Kumar et al., 2016): A real-world signed network with $5,881$ nodes and $35,592$ edges, weighted from $-10$ to $10$, capturing trust relationships among Bitcoin users.

The construction of the training and testing datasets is summarized in Table 1. The training set consists of 500 3-regular, 500 5-regular graphs, and 500 7-regular graphs with 100 nodes each, corresponding to the cases $k = 2$, $k = 3$, and $k = 10$ respectively. The test set of random regular graphs includes 20 3-regular and 20 5-regular graphs for each $k \in \{2, 3\}$, with node counts of 100, 1,000, and 10,000. For the Gset benchmark, we evaluate both unweighted and weighted variants. The unweighted test set includes all Gset instances, with results reported in Tables 6 and 7 in Appendix D. For the weighted variant, we generate perturbations of the four largest Gset graphs (G70, G72, G77, G81) by multiplying each edge weight by $\sigma \sim \mathcal{U}[l, u]$, creating 10 perturbed instances per graph. We examine three distinct perturbation regimes: (i) mild perturbations ($[0.9, 1.1]$), (ii) moderate variations ($[0, 10]$), and (iii) extreme modifications ($[0, 100]$). The moderate perturbation results ($[0, 10]$) are presented in Table 3, with the remaining cases available in Appendix E. Additionally, we evaluate performance on three COLOR benchmark instances: `anna`, `david`, and `huck`.

**Model Settings.** `ROS` is designed as a two-layer GNN, with both the input and hidden dimensions set to 100. To address

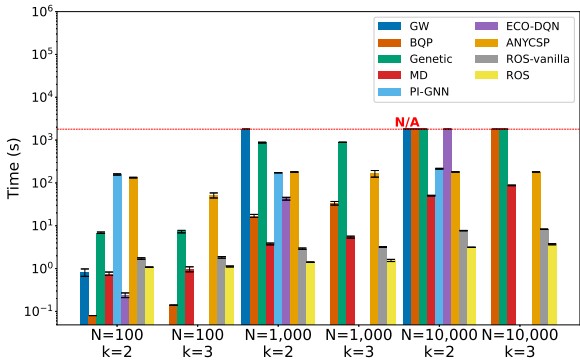

(a) Random regular graph

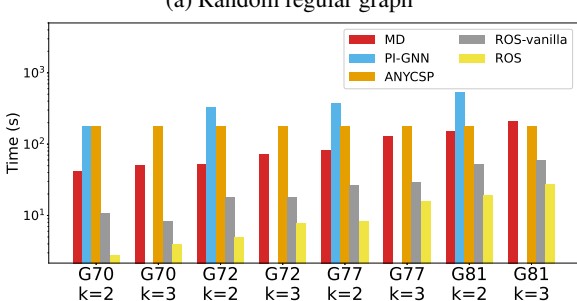

(b) Weighted Gset with perturbation ratio $[0, 10]$

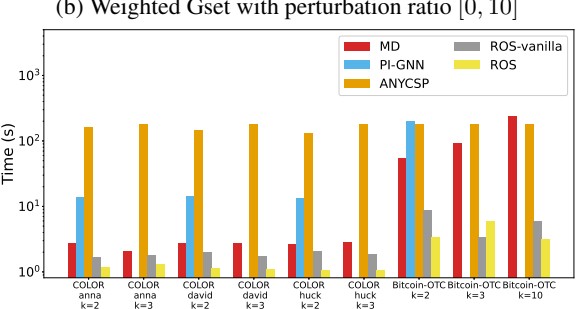

(c) COLOR datasets and Bitcoin-OTC datasets

Figure 2: The computational time comparison of Max-$k$-Cut problems.

the issue of gradient vanishing, we apply graph normalization as proposed by Cai et al. (2021). The `ROS` model is pre-training using Adam with a learning rate of $10^{-2}$ for one epoch. During fine-tuning, the model is further optimized using the same Adam optimizer and learning rate, applying early stopping with a tolerance of $10^{-2}$ and patience of 100 iterations. Training terminates if no improvement is observed. Finally, in the random sampling stage, we execute Algorithm 1 for $T = 100$ trials and return the best solution.

**Evaluation Configuration.** All our experiments were conducted on an NVIDIA RTX 3090 GPU, using PyTorch 2.2.0.

Table 1: Statistics of the training and testing datasets.

| | Dataset | Graph Type | $N$ | # Graphs | Weight Type |
|---|---|---|---|---|---|
| Train | Random Regular Graph | regular | 100 | 500 | unweighted |
| Test | Random Regular Graph | regular | 100, 1,000, 10,000 | 60 | unweighted |
| | Gset | random, planar, toroidal | $800 \sim 20,000$ | 71 | unweighted, weighted |
| | COLOR | real-world | 74, 87, 138 | 3 | unweighted |
| | Bitcoin-OTC | real-world | 5,881 | 1 | weighted |

Table 2: Cut value comparison of Max-$k$-Cut problems on random regular graphs.

| Methods | N=100 | | N=1,000 | | N=10,000 | |
|---|---|---|---|---|---|---|
| | $k=2$ | $k=3$ | $k=2$ | $k=3$ | $k=2$ | $k=3$ |
| GW | $130.20_{\pm2.79}$ | – | N/A | – | N/A | – |
| BQP | $131.55_{\pm2.42}$ | $239.70_{\pm1.82}$ | $1324.45_{\pm6.34}$ | $2419.15_{\pm6.78}$ | N/A | N/A |
| Genetic | $127.55_{\pm2.82}$ | $235.50_{\pm3.15}$ | $1136.65_{\pm10.37}$ | $2130.30_{\pm8.49}$ | N/A | N/A |
| MD | $127.20_{\pm2.16}$ | $235.50_{\pm3.29}$ | $1250.35_{\pm11.21}$ | $2344.85_{\pm9.86}$ | $12428.85_{\pm26.13}$ | $23341.20_{\pm32.87}$ |
| PI-GNN | $122.95_{\pm3.83}$ | – | $1210.45_{\pm44.56}$ | – | $12655.05_{\pm94.25}$ | – |
| ECO-DQN | $135.60_{\pm1.53}$ | – | $1366.20_{\pm5.20}$ | – | N/A | – |
| ANYCSP | $131.65_{\pm3.35}$ | $247.90_{\pm0.89}$ | $1366.05_{\pm5.25}$ | $2494.50_{\pm2.99}$ | $13692.35_{\pm11.27}$ | $24929.80_{\pm7.53}$ |
| ROS-vanilla | $132.80_{\pm1.99}$ | $243.20_{\pm1.80}$ | $1322.95_{\pm6.57}$ | $2443.9_{\pm4.10}$ | $13239.80_{\pm14.71}$ | $24413.30_{\pm16.02}$ |
| ROS | $128.20_{\pm2.82}$ | $240.30_{\pm2.59}$ | $1283.75_{\pm6.89}$ | $2405.75_{\pm5.72}$ | $12856.85_{\pm26.50}$ | $24085.95_{\pm21.88}$ |

## 4.2. Performance Comparison against Baselines

### 4.2.1. COMPUTATIONAL TIME

We evaluated ROS against seven baseline algorithms: GW, BQP, Genetic, MD, PI-GNN, ECO-DQN, and ANYCSP on random regular graphs, comparing computational time for both Max-Cut and Max-3-Cut tasks. Experiments covered three problem scales: $N = 100$, $N = 1,000$, and $N = 10,000$, with results shown in Figure 2a. For larger instances, Figure 2b compares the scalable methods (MD, ANYCSP, and PI-GNN) on weighted Gset graphs ($N \geq 10,000$) with edge weight perturbations in $[0, 10]$. Figure 2c extends this comparison to real-world networks (COLOR and Bitcoin-OTC graphs). Instances marked "N/A" indicate timeout failures (30-minute limit). Complete results for unweighted Gset benchmarks, including comparisons with state-of-the-art methods LPI and MOH, are provided in Tables 6 and 7 (Appendix D).

The results depicted in Figure 2a indicate that ROS efficiently solves all problem instances within seconds, even for large problem sizes of $N = 10,000$. In terms of baseline performance, the approximation algorithm GW performs efficiently on instances with $N = 100$, but it struggles with larger sizes due to the substantial computational burden associated with solving the underlying semi-definite programming problem. Heuristic methods such as BQP and Genetic can manage cases up to $N = 1,000$ in a few hundred seconds, yet they fail to solve larger instances with $N = 10,000$ because of the high computational cost of

each iteration. Notably, MD is the only traditional method capable of solving large instances within a reasonable time frame; however, when $N$ reaches $10,000$, the computational time for MD approaches 15 times that of ROS. Regarding L2O methods, PI-GNN necessitates retraining and prediction for each instance, with test times exceeding dozens of seconds even for $N = 100$. ECO-DQN relies on expensive GNNs at each decision step and can not scale to large problem sizes of $N = 10,000$. ANYCSP needs hundreds of seconds even for $N = 100$ due to the global search operation and long sampling trajectory. In contrast, ROS solves these large instances in merely a few seconds throughout the experiments, requiring only $10\%$ of the computational time utilized by other L2O baselines. Figure 2b and Figure 2c illustrate the results for the weighted Gset benchmark and real-world datasets, respectively, where ROS efficiently solves the largest instances in just a few seconds, while other methods take tens to hundreds of seconds for equivalent tasks. Remarkably, ROS utilizes only about $1\%$ of the computational time required by PI-GNN.

### 4.2.2. CUT VALUE

We evaluate ROS's performance on random regular graphs, the Gset benchmark, and real-world datasets, measuring solution quality for Problem (1). Results appear in Tables 2 (random graphs), 3 (weighted Gset), and 4 (real-world data), where "–" denotes methods incompatible with Max-$k$-Cut problems.

Table 3: Cut value comparison of Max-$k$-Cut problems on weighted Gset instances, where the noise factor $\sigma \sim [0, 10]$.

| Methods | G70 (N=10,000) | | G72 (N=10,000) | | G77 (N=14,000) | | G81 (N=20,000) | |
|---|---|---|---|---|---|---|---|---|
| | $k = 2$ | $k = 3$ | $k = 2$ | $k = 3$ | $k = 2$ | $k = 3$ | $k = 2$ | $k = 3$ |
| GW | N/A | – | N/A | – | N/A | – | N/A | – |
| BQP | N/A | N/A | N/A | N/A | N/A | N/A | N/A | N/A |
| Genetic | N/A | N/A | N/A | N/A | N/A | N/A | N/A | N/A |
| MD | 45490.21 | 49615.85 | 33449.49 | 38798.78 | 47671.94 | 55147.26 | 67403.00 | 78065.07 |
| PI-GNN | 44275.72 | – | 31469.65 | – | 44359.72 | – | 62439.97 | – |
| ECO-DQN | N/A | – | N/A | – | N/A | – | N/A | – |
| ANYCSP | 46420.48 | 48831.32 | $-280.74$ | $-208.01$ | 845.72 | 988.96 | $-13.52$ | 271.01 |
| ROS-vanilla | 47140.07 | 49826.90 | 36697.11 | 42067.80 | 52226.53 | 59636.36 | 74051.42 | 84498.44 |
| ROS | 46707.60 | 49813.45 | 35733.11 | 40987.92 | 50790.44 | 58253.31 | 72057.24 | 82450.68 |

Table 4: Cut value comparison of Max-$k$-Cut problems on COLOR datasets and Bitcoin-OTC Datasets.

| Methods | COLOR anna | | COLOR david | | COLOR huck | | Bitcoin-OTC | | |
|---|---|---|---|---|---|---|---|---|---|
| | $k = 2$ | $k = 3$ | $k = 2$ | $k = 3$ | $k = 2$ | $k = 3$ | $k = 2$ | $k = 3$ | $k = 10$ |
| MD | 339 | 421 | 259 | 329 | 184 | 242 | 39076 | 47595 | 53563 |
| PI-GNN | 279 | – | 228 | – | 166 | – | 37216 | – | – |
| ANYCSP | 330 | 423 | 263 | 328 | 166 | 139 | 10431 | 14265 | 19372 |
| ROS-vanilla | 351 | 429 | 266 | 336 | 191 | 246 | 40576 | 48214 | 53758 |
| ROS | 351 | 423 | 266 | 324 | 191 | 242 | 39850 | 48980 | 53778 |

The results demonstrate that ROS consistently produces high-quality solutions for both $k = 2$ and $k = 3$ across all scales. While GW performs well for Max-Cut ($k = 2$) at $N = 100$, it fails to generalize to arbitrary $k$. Traditional methods like BQP and Genetic support $k = 3$ but often converge to suboptimal solutions. Although MD handles general $k$, it consistently underperforms ROS. Among learning-based methods, PI-GNN proves unsuitable for $k = 3$ due to QUBO incompatibility and unreliable heuristic rounding, while ECO-DQN lacks $k = 3$ support entirely. While ANYCSP achieves strong results on unweighted graphs, it cannot process weighted instances. These experiments collectively show that ROS offers superior generalizability and robustness for weighted Max-$k$-Cut tasks, outperforming both traditional and learning-based approaches in solution quality and flexibility.

To further assess ROS's scalability, we conduct comprehensive benchmarking against scalable baselines using challenging real-world datasets, including the COLOR and Bitcoin-OTC networks. The results in Table 4 demonstrate that both ROS and its simplified variant ROS-vanilla consistently outperform competing methods across most experimental settings. This performance advantage is particularly pronounced for the weighted Bitcoin-OTC instances, where our approach achieves superior solution quality while maintaining computational efficiency.

### 4.3. Effect of the "Pre-train" Stage in ROS

To evaluate the impact of the pre-training stage in ROS, we compared it with ROS-vanilla, which omits pre-training (see Section 3.3). We assessed both methods based on cut values and computational time. Figure 3 illustrates the ratios of these metrics between ROS-vanilla and ROS. In this figure, the horizontal axis represents the problem instances, while the left vertical axis (green bars) displays the ratio of objective function values, and the right vertical axis (red curve) indicates the ratio of computational times.

As shown in Figure 3a, ROS-vanilla achieves higher objective function values in most settings on the random regular graphs; however, its computational time is approximately 1.5 times greater than that of ROS. Thus, ROS demonstrates a faster solving speed compared to ROS-vanilla. Similarly, in experiments conducted on the Gset benchmark (Figure 3b), ROS reduces computational time by around 40% while maintaining performance comparable to that of ROS-vanilla. Notably, in the Max-3-Cut problem for the largest instance, G81, ROS effectively halves the solving time, showcasing the significant acceleration effect of pre-training. It is worth mentioning that the ROS model was pre-trained on random regular graphs with $N = 100$ and generalized well to regular graphs with $N = 1,000$ and $N = 10,000$, as well as to Gset problem instances of varying sizes and types. This illustrates ROS's capability to

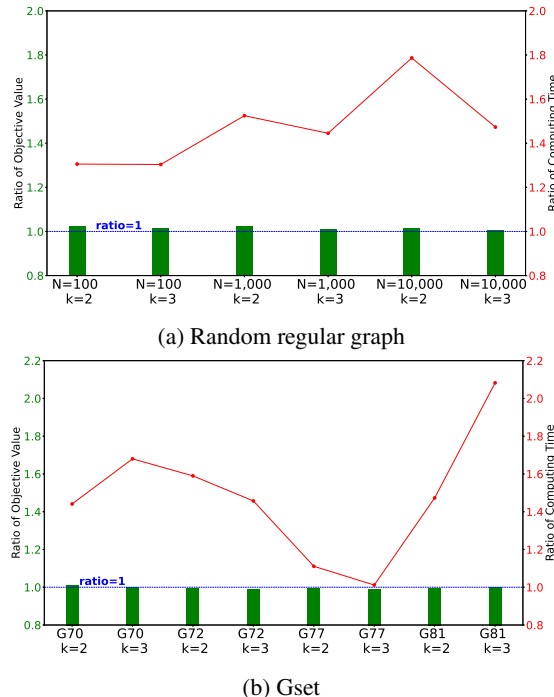

(a) Random regular graph

(b) Gset

Figure 3: The ratio of computational time and cut value comparison between `ROS-vanilla` and `ROS`.

generalize and accelerate the solving of large-scale problems across diverse graph types and sizes, emphasizing the strong out-of-distribution generalization afforded by pre-training.

In summary, while `ROS-vanilla` achieves slightly higher objective function values on individual instances, it requires longer solving times and struggles to generalize to other problem instances. This observation highlights the trade-off between a model's ability to generalize and its capacity to fit specific instances. Specifically, a model that fits individual instances exceptionally well may fail to generalize to new data, resulting in longer solving times. Conversely, a model that generalizes effectively may exhibit slightly weaker performance on specific instances, leading to a marginal decrease in objective function values. Therefore, the choice between these two training modes should be guided by the specific requirements of the application.

## 5. Conclusions

In this paper, we propose ROS, an efficient method for addressing the Max-$k$-Cut problem with arbitrary edge weights. Our approach begins by relaxing the constraints of the original discrete problem to probabilistic simplices. To effectively solve this relaxed problem, we propose an optimization algorithm based on GNN parametrization and incorporate transfer learning by leveraging pre-trained GNNs to warm-start the training process. After resolving the re-

laxed problem, we present a novel random sampling algorithm that maps the continuous solution back to a discrete form. By integrating geometric landscape analysis with statistical theory, we establish the consistency of function values between the continuous and discrete solutions. Experiments conducted on random regular graphs, the Gset benchmark, and real-world datasets demonstrate that our method is highly efficient for solving large-scale Max-$k$-Cut problems, requiring only a few seconds, even for instances with tens of thousands of variables. Furthermore, it exhibits robust generalization capabilities across both in-distribution and out-of-distribution instances, highlighting its effectiveness for large-scale optimization tasks. Exploring other sampling algorithms to further boost ROS performance is a future research direction. Moreover, the ROS framework with theoretical insights could be potentially extended to other graph-related combinatorial problems, and this direction is also worth investigating as future work.

## Impact Statement

This paper presents work whose goal is to advance the field of Machine Learning. There are many potential societal consequences of our work, none of which we feel must be specifically highlighted here.

## Acknowledgement

This work was supported by the National Key R&D Program of China under grant 2022YFA1003900. Ye Xue acknowledges support from the National Natural Science Foundation of China (Grant No. 62301334), the Guangdong Major Project of Basic and Applied Basic Research (No. 2023B0303000001), Akang Wang also acknowledges support from the National Natural Science Foundation of China (Grant No. 12301416), the Guangdong Basic and Applied Basic Research Foundation (Grant No. 2024A1515010306).

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

## A. Related Works

Relaxation-based methods have been central to the algorithmic design for Max-Cut and its generalizations. In Table 5, we compare our proposed probability simplex relaxation with several representative approaches along key dimensions: variable complexity (# Var.), applicability to general Max-$k$-Cut, polynomial-time solvability, objective value consistency with the original problem, and scalability to large instances.

Table 5: Comparison between Different Relaxations

| Relaxation | # Var. | Max-$k$-Cut | Polynomial Solvable? | Obj. Value Consistency? | Scalable? |
|---|---|---|---|---|---|
| Lovasz Extension (Bach, 2013) | $\mathcal{O}(N)$ | ✗ | ✗ | ✓ | ✓ |
| SDP Relaxation (Goemans & Williamson, 1995) | $\mathcal{O}(N^2)$ | ✗ | ✓ | ✗ | ✗ |
| SDP Relaxation (Frieze & Jerrum, 1997) | $\mathcal{O}(N \times k)$ | ✓ | ✓ | ✗ | ✗ |
| Rank-2 Relaxation (Burer et al., 2002) | $\mathcal{O}(N)$ | ✗ | ✗ | ✗ | ✓ |
| QUBO Relaxation (Andrade et al., 2024) | $\mathcal{O}(N)$ | ✗ | ✗ | ✗ | ✓ |
| Probability Simplex Relaxation (ours) | $\mathcal{O}(N \times k)$ | ✓ | ✗ | ✓ | ✓ |

The Lovász extension (Bach, 2013), originally designed for submodular optimization, admits scalable convex formulations but does not extend naturally to general Max-$k$-Cut problems. Seminal SDP-based methods, such as Goemans-Williamson (Goemans & Williamson, 1995) for Max-Cut and its $k$-way extension (Frieze & Jerrum, 1997), offer polynomial-time approximation guarantees. However, their reliance on large-scale semidefinite programming limits practical scalability and makes them less effective on modern large-scale instances. Non-convex formulations, including the rank-2 relaxation (Burer et al., 2002) and QUBO-based relaxation (Andrade et al., 2024), provide scalable alternatives for Max-Cut but lack theoretical guarantees for Max-$k$-Cut and are typically solved locally. These methods often exhibit poor objective consistency and limited generalization.

In contrast, our probability simplex relaxation introduces a non-convex yet tractable formulation for Max-$k$-Cut with $\mathcal{O}(N \times k)$ variables. While it is not globally solvable in polynomial time, its optimal value aligns exactly with that of the original Max-$k$-Cut problem. Empirically, our GNN-based solver produces high-quality fractional solutions, which serve as effective initializations for randomized sampling. Overall, the proposed relaxation strikes a favorable balance between expressiveness, consistency, and scalability, offering a practical and theoretically grounded solution framework for large-scale Max-$k$-Cut problems.

## B. Proof of Theorem 3.2

*Proof.* Before proceeding with the proof of Theorem 3.2, we first define the neighborhood of a vector $\bar{\boldsymbol{x}} \in \Delta_k$, and establish results of Lemma B.2 and Lemma B.3.

**Definition B.1.** Let $\bar{\boldsymbol{x}} = (\bar{\boldsymbol{x}}_1, \cdots, \bar{\boldsymbol{x}}_k)$ denote a point in $\Delta_k$. We define the neighborhood induced by $\bar{\boldsymbol{x}}$ as follows:

$$\widetilde{\mathcal{N}}(\bar{\boldsymbol{x}}) := \left\{ (\boldsymbol{x}_1, \cdots, \boldsymbol{x}_k) \in \Delta_k \ \middle| \ \sum_{j \in \mathcal{K}(\bar{\boldsymbol{x}})} \boldsymbol{x}_j = 1 \right\},$$

where $\mathcal{K}(\bar{\boldsymbol{x}}) = \{j \in \{1, \cdots, k\} \mid \bar{\boldsymbol{x}}_j > 0\}$.

**Lemma B.2.** *Given $\boldsymbol{X}_{\cdot i} \in \widetilde{\mathcal{N}}(\boldsymbol{X}^{\star}_{\cdot i})$, it follows that*

$$\mathcal{K}(\boldsymbol{X}_{\cdot i}) \subseteq \mathcal{K}(\boldsymbol{X}^{\star}_{\cdot i}).$$

*Proof.* Suppose there exists $j \in \mathcal{K}(\boldsymbol{X}_{\cdot i})$ such that $j \notin \mathcal{K}(\boldsymbol{X}^{\star}_{\cdot i})$, implying $\boldsymbol{X}_{ji} > 0$ and $\boldsymbol{X}^{\star}_{ji} = 0$.

We then have

$$\sum_{l \in \mathcal{K}(\boldsymbol{X}^{\star}_{\cdot i})} \boldsymbol{X}_{li} + \boldsymbol{X}_{ji} \leq \sum_{l=1}^{k} \boldsymbol{X}_{li} = 1,$$

which leads to

$$\sum_{l \in \mathcal{K}(\boldsymbol{X}_{\cdot i}^\star)} \boldsymbol{X}_{li} \leq 1 - \boldsymbol{X}_{ji} < 1,$$

contradicting with the fact that $\boldsymbol{X}_{\cdot i} \in \widetilde{\mathcal{N}}(\boldsymbol{X}_{\cdot i}^\star)$. $\qquad\square$

**Lemma B.3.** *Let $\boldsymbol{X}^\star$ be a globally optimal solution to $\bar{\boldsymbol{P}}$, then*

$$f(\boldsymbol{X}; \boldsymbol{W}) = f(\boldsymbol{X}^\star; \boldsymbol{W}),$$

*where $\boldsymbol{X}$ has only the $i^{th}$ column $\boldsymbol{X}_{\cdot i} \in \widetilde{\mathcal{N}}(\boldsymbol{X}_{\cdot i}^\star)$, and other columns are identical to those of $\boldsymbol{X}^\star$. Moreover, $\boldsymbol{X}$ is also a globally optimal solution to $\bar{\boldsymbol{P}}$.*

*Proof.* The fact that $\boldsymbol{X}$ is a globally optimal solution to $\bar{\boldsymbol{P}}$ follows directly from the equality $f(\boldsymbol{X}; \boldsymbol{W}) = f(\boldsymbol{X}^\star; \boldsymbol{W})$. Thus, it suffices to prove this equality. Consider that $\boldsymbol{X}^\star$ and $\boldsymbol{X}$ differ only in the $i^{th}$ column, and $\boldsymbol{X}_{\cdot i} \in \widetilde{\mathcal{N}}(\boldsymbol{X}_{\cdot i}^\star)$. We can rewrite the objective value function as

$$f(\boldsymbol{X}; \boldsymbol{W}) = g(\boldsymbol{X}_{\cdot i}; \boldsymbol{X}_{\cdot -i}) + h(\boldsymbol{X}_{\cdot -i}),$$

where $\boldsymbol{X}_{\cdot -i}$ represents all column vectors of $\boldsymbol{X}$ except the $i^{th}$ column. The functions $g$ and $h$ are defined as follows:

$$g(\boldsymbol{X}_{\cdot i}; \boldsymbol{X}_{\cdot -i}) = \sum_{j=1}^{N} \boldsymbol{W}_{ij} \boldsymbol{X}_{\cdot i}^\top \boldsymbol{X}_{\cdot j} + \sum_{j=1}^{N} \boldsymbol{W}_{ji} \boldsymbol{X}_{\cdot j}^\top \boldsymbol{X}_{\cdot i} - \boldsymbol{W}_{ii} \boldsymbol{X}_{\cdot i}^\top \boldsymbol{X}_{\cdot i},$$

$$h(\boldsymbol{X}_{\cdot -i}) = \sum_{l=1, l \neq i}^{N} \sum_{j=1, j \neq i}^{N} \boldsymbol{W}_{lj} \boldsymbol{X}_{\cdot l}^\top \boldsymbol{X}_{\cdot j}$$

To establish that $f(\boldsymbol{X}; \boldsymbol{W}) = f(\boldsymbol{X}^\star; \boldsymbol{W})$, it suffices to show that

$$g(\boldsymbol{X}_{\cdot i}; \boldsymbol{X}_{\cdot -i}) = g(\boldsymbol{X}_{\cdot i}^\star; \boldsymbol{X}_{\cdot -i})$$

as $\boldsymbol{X}_{\cdot -i} = \boldsymbol{X}_{\cdot -i}^\star$.

Rewriting $g(\boldsymbol{X}_{\cdot i}; \boldsymbol{X}_{\cdot -i})$, we obtain

$$\begin{aligned}
g(\boldsymbol{X}_{\cdot i}; \boldsymbol{X}_{\cdot -i}) &= \sum_{j=1}^{N} \boldsymbol{W}_{ij} \boldsymbol{X}_{\cdot i}^\top \boldsymbol{X}_{\cdot j} + \sum_{j=1}^{N} \boldsymbol{W}_{ji} \boldsymbol{X}_{\cdot j}^\top \boldsymbol{X}_{\cdot i} \\
&= 2 \sum_{j=1}^{N} \boldsymbol{W}_{ij} \boldsymbol{X}_{\cdot i}^\top \boldsymbol{X}_{\cdot j} \\
&= 2 \boldsymbol{X}_{\cdot i}^\top \sum_{j=1, j \neq i}^{N} \boldsymbol{W}_{ij} \boldsymbol{X}_{\cdot j} \\
&= 2 \boldsymbol{X}_{\cdot i}^\top \boldsymbol{Y}_{\cdot i},
\end{aligned}$$

where $\boldsymbol{Y}_{\cdot i} := \sum_{j=1, j \neq i}^{N} \boldsymbol{W}_{ij} \boldsymbol{X}_{\cdot j}$.

If $|\mathcal{K}(\boldsymbol{X}_{\cdot i}^\star)| = 1$, then there is only one non-zero element in $\boldsymbol{X}_{\cdot i}^\star$ equal to one. Therefore, $g(\boldsymbol{X}_{\cdot i}^\star; \boldsymbol{X}_{\cdot -i}) = g(\boldsymbol{X}_{\cdot i}; \boldsymbol{X}_{\cdot -i})$ since $\boldsymbol{X}_{\cdot i} = \boldsymbol{X}_{\cdot i}^\star$.

For the case where $|\mathcal{K}(\boldsymbol{X}_{\cdot i}^\star)| > 1$, we consider any indices $j, l \in \mathcal{K}(\boldsymbol{X}_{\cdot i}^\star)$ such that $\boldsymbol{X}_{ji}^\star, \boldsymbol{X}_{li}^\star \in (0, 1)$. Then, there exists $\epsilon > 0$ such that we can construct a point $\widetilde{\boldsymbol{x}} \in \Delta_k$ where the $j^{th}$ element is set to $\boldsymbol{X}_{ji}^\star - \epsilon$, the $l^{th}$ element is set to $\boldsymbol{X}_{li}^\star + \epsilon$, and all other elements remain the same as in $\boldsymbol{X}_{\cdot i}^\star$. Since $\boldsymbol{X}^\star$ is a globally optimum of the function $f(\boldsymbol{X}; \boldsymbol{W})$, it follows that $\boldsymbol{X}_{\cdot i}^\star$ is also a global optimum for the function $g(\boldsymbol{X}_{\cdot i}^\star; \boldsymbol{X}_{\cdot -i})$. Thus, we have

$$\begin{aligned}
g(\boldsymbol{X}_{\cdot i}^\star; \boldsymbol{X}_{\cdot -i}) &\leq g(\widetilde{\boldsymbol{x}}; \boldsymbol{X}_{\cdot -i}) \\
\boldsymbol{X}_{\cdot i}^{\star\top} \boldsymbol{Y}_{\cdot i} &\leq \widetilde{\boldsymbol{x}}^\top \boldsymbol{Y}_{\cdot i} \\
&= \boldsymbol{X}_{\cdot i}^{\star\top} \boldsymbol{Y}_{\cdot i} - \epsilon \boldsymbol{Y}_{ji} + \epsilon \boldsymbol{Y}_{li},
\end{aligned}$$

which leads to the inequality

$$\boldsymbol{Y}_{ji} \leq \boldsymbol{Y}_{li}. \tag{3}$$

Next, we can similarly construct another point $\hat{\boldsymbol{x}} \in \Delta_k$ with its $j^{th}$ element equal to $\boldsymbol{X}_{ji}^{\star} + \epsilon$, the $k^{th}$ element equal to $\boldsymbol{X}_{ki}^{\star} - \epsilon$, and all other elements remain the same as in $\boldsymbol{X}_{\cdot i}^{\star}$. Subsequently, we can also derive that

$$
\begin{aligned}
g(\boldsymbol{X}_{\cdot i}^{\star}; \boldsymbol{X}_{\cdot -i}) &\leq g(\hat{\boldsymbol{x}}; \boldsymbol{X}_{\cdot -i}) \\
&= \boldsymbol{X}_{\cdot i}^{\star\top} \boldsymbol{Y}_{\cdot i} + \epsilon \boldsymbol{Y}_{ji} - \epsilon \boldsymbol{Y}_{li},
\end{aligned}
$$

which leads to another inequality

$$\boldsymbol{Y}_{li} \leq \boldsymbol{Y}_{ji}. \tag{4}$$

Consequently, combined inequalities (3) and (4), we have

$$\boldsymbol{Y}_{ji} = \boldsymbol{Y}_{li},$$

for $j, l \in \mathcal{K}(\boldsymbol{X}_{\cdot i}^{\star})$.

From this, we can deduce that

$$\boldsymbol{Y}_{j_1 i} = \boldsymbol{Y}_{j_2 i} = \cdots = \boldsymbol{Y}_{j_{|\mathcal{K}(\boldsymbol{X}_{\cdot i}^{\star})|} i} = t,$$

where $j_1, \cdots, j_{|\mathcal{K}(\boldsymbol{X}_{\cdot i}^{\star})|} \in \mathcal{K}(\boldsymbol{X}_{\cdot i}^{\star})$.

Next, we find that

$$
\begin{aligned}
g(\boldsymbol{X}_{\cdot i}^{\star}; \boldsymbol{X}_{\cdot -i}) &= 2 \boldsymbol{X}_{\cdot i}^{\star\top} \boldsymbol{Y}_{\cdot i} \\
&= 2 \sum_{j=1}^{k} \boldsymbol{X}_{ji}^{\star} \boldsymbol{Y}_{ji} \\
&= 2 \sum_{j=1, j \in \mathcal{K}(\boldsymbol{X}_{\cdot i}^{\star})}^{N} \boldsymbol{X}_{ji}^{\star} \boldsymbol{Y}_{ji} \\
&= 2t \sum_{j=1, j \in \mathcal{K}(\boldsymbol{X}_{\cdot i}^{\star})}^{N} \boldsymbol{X}_{ji}^{\star} \\
&= 2t.
\end{aligned}
$$

Similarly, we have

$$
\begin{aligned}
g(\boldsymbol{X}_{\cdot i}; \boldsymbol{X}_{\cdot -i}) &= 2 \boldsymbol{X}_{\cdot i}^{\top} \boldsymbol{Y}_{\cdot i} \\
&= 2 \sum_{j=1}^{k} \boldsymbol{X}_{ji} \boldsymbol{Y}_{ji} \\
&= 2 \sum_{j=1, j \in \mathcal{K}(\boldsymbol{X}_{\cdot i})} \boldsymbol{X}_{ji} \boldsymbol{Y}_{ji} \\
&\overset{\text{Lemma B.2}}{=} 2t \sum_{j=1, j \in \mathcal{K}(\boldsymbol{X}_{\cdot i})} \boldsymbol{X}_{ji} \\
&= 2t
\end{aligned}
$$

Accordingly, we conclude that

$$g(\boldsymbol{X}_{\cdot i}; \boldsymbol{X}_{\cdot -i}) = g(\boldsymbol{X}_{\cdot i}^{\star}; \boldsymbol{X}_{\cdot -i}),$$

which leads us to the result

$$f(\boldsymbol{X}; \boldsymbol{W}) = f(\boldsymbol{X}^{\star}; \boldsymbol{W}),$$

where $\boldsymbol{X}_{\cdot i} \in \widetilde{\mathcal{N}}(\boldsymbol{X}_{\cdot i}^{\star})$, $\boldsymbol{X}_{\cdot -i} = \boldsymbol{X}_{\cdot -i}^{\star}$. $\qquad\square$

Accordingly, for any $\boldsymbol{X} \in \mathcal{N}(\boldsymbol{X}^\star)$, we iteratively apply Lemma B.3 to each column of $\boldsymbol{X}^\star$ while holding the other columns fixed, thereby proving Theorem 3.2.

$\square$

## C. Proof of Theorem 3.3

*Proof.* Based on $\overline{\boldsymbol{X}}$, we can construct the random variable $\widetilde{\boldsymbol{X}}$, where $\widetilde{\boldsymbol{X}}_{\cdot i} \sim \mathrm{Cat}(\boldsymbol{x}; \boldsymbol{p} = \overline{\boldsymbol{X}}_{\cdot i})$. The probability mass function is given by

$$\mathbf{P}(\widetilde{\boldsymbol{X}}_{\cdot i} = \boldsymbol{e}_\ell) = \overline{\boldsymbol{X}}_{\ell i}, \tag{5}$$

where $\ell = 1, \cdots, k$.

Next, we have

$$\mathbb{E}_{\widetilde{\boldsymbol{X}}}[f(\widetilde{\boldsymbol{X}}; \boldsymbol{W})] = \mathbb{E}_{\widetilde{\boldsymbol{X}}}[\widetilde{\boldsymbol{X}} \boldsymbol{W} \widetilde{\boldsymbol{X}}^\top] = \mathbb{E}_{\widetilde{\boldsymbol{X}}}[\sum_{i=1}^{N} \sum_{j=1}^{N} \boldsymbol{W}_{ij} \widetilde{\boldsymbol{X}}_{\cdot i}^\top \widetilde{\boldsymbol{X}}_{\cdot j}]$$

$$= \sum_{i=1}^{N} \sum_{j=1}^{N} \boldsymbol{W}_{ij} \mathbb{E}_{\widetilde{\boldsymbol{X}}_{\cdot i} \widetilde{\boldsymbol{X}}_{\cdot j}}[\widetilde{\boldsymbol{X}}_{\cdot i}^\top \widetilde{\boldsymbol{X}}_{\cdot j}]$$

$$= \sum_{i=1}^{N} \sum_{j=1}^{N} \boldsymbol{W}_{ij} \mathbb{E}_{\widetilde{\boldsymbol{X}}_{\cdot i} \widetilde{\boldsymbol{X}}_{\cdot j}}[\mathbb{1}(\widetilde{\boldsymbol{X}}_{\cdot i} = \widetilde{\boldsymbol{X}}_{\cdot j})]$$

$$= \sum_{i=1}^{N} \sum_{j=1}^{N} \boldsymbol{W}_{ij} \mathbb{P}(\widetilde{\boldsymbol{X}}_{\cdot i} = \widetilde{\boldsymbol{X}}_{\cdot j})$$

$$= \sum_{i=1}^{N} \sum_{j=1, j \neq i}^{N} \boldsymbol{W}_{ij} \mathbb{P}(\widetilde{\boldsymbol{X}}_{\cdot i} = \widetilde{\boldsymbol{X}}_{\cdot j}). \tag{6}$$

Since $\widetilde{\boldsymbol{X}}_{\cdot i}$ and $\widetilde{\boldsymbol{X}}_{\cdot j}$ are independent for $i \neq j$, we have

$$\mathbb{P}(\widetilde{\boldsymbol{X}}_{\cdot i} = \widetilde{\boldsymbol{X}}_{\cdot j}) = \sum_{\ell=1}^{k} \mathbb{P}(\widetilde{\boldsymbol{X}}_{\cdot i} = \widetilde{\boldsymbol{X}}_{\cdot j} = \boldsymbol{e}_\ell)$$

$$= \sum_{\ell=1}^{k} \mathbb{P}(\widetilde{\boldsymbol{X}}_{\cdot i} = \boldsymbol{e}_\ell, \widetilde{\boldsymbol{X}}_{\cdot j} = \boldsymbol{e}_\ell)$$

$$= \sum_{\ell=1}^{k} \mathbb{P}(\widetilde{\boldsymbol{X}}_{\cdot i} = \boldsymbol{e}_\ell) \mathbb{P}(\widetilde{\boldsymbol{X}}_{\cdot j} = \boldsymbol{e}_\ell)$$

$$= \sum_{\ell=1}^{k} \overline{\boldsymbol{X}}_{\ell i} \overline{\boldsymbol{X}}_{\ell j}$$

$$= \overline{\boldsymbol{X}}_{\cdot i}^\top \overline{\boldsymbol{X}}_{\cdot j}. \tag{7}$$

Substitute (7) into (6), we obtain

$$\mathbb{E}_{\widetilde{\boldsymbol{X}}}[f(\widetilde{\boldsymbol{X}}; \boldsymbol{W})] = \sum_{i=1}^{N} \sum_{j=1}^{N} \boldsymbol{W}_{ij} \overline{\boldsymbol{X}}_{\cdot i}^\top \overline{\boldsymbol{X}}_{\cdot j} = f(\overline{\boldsymbol{X}}; \boldsymbol{W}). \tag{8}$$

$\square$

## D. The Results on Unweighted Gset Instances

Table 6: Complete results on Gset instances for Max-Cut.

| Instance | $|\mathcal{V}|$ | $|\mathcal{E}|$ | GW Obj.↑ | GW Time (s)↓ | MD Obj.↑ | MD Time (s)↓ | PI-GNN Obj.↑ | PI-GNN Time (s)↓ | Genetic Obj.↑ | Genetic Time (s)↓ | BQP Obj.↑ | BQP Time (s)↓ | ECO-DQN Obj.↑ | ECO-DQN Time (s)↓ | ANYCSP Obj.↑ | ANYCSP Time (s)↓ | MOH Obj.↑ | MOH Time (s)↓ | LPI Obj.↑ | LPI Time (s)↓ | ROS-vanilla Obj.↑ | ROS-vanilla Time (s)↓ | ROS Obj.↑ | ROS Time (s)↓ |
|---|---|---|---|---|---|---|---|---|---|---|---|---|---|---|---|---|---|---|---|---|---|---|---|---|
| G1 | 800 | 19176 | 11299 | 1228.0 | 11320 | 5.1 | 10680 | 214.3 | 10929 | 587.4 | 11406 | 11.3 | 11482 | 23.4 | 11574 | 180.1 | 11624 | 1.5 | 11624 | 7 | 11423 | 2.6 | 11395 | 1.7 |
| G2 | 800 | 19176 | 11299 | 1225.4 | 11255 | 5.3 | 10533 | 212.7 | 10926 | 588.3 | 11426 | 11.7 | 11516 | 25.2 | 11591 | 180.1 | 11620 | 4.6 | 11620 | 8 | 11462 | 2.6 | 11467 | 1.8 |
| G3 | 800 | 19176 | 11289 | 1243.2 | 11222 | 5.3 | 10532 | 215.1 | 10933 | 596.8 | 11397 | 11.0 | 11543 | 26.1 | 11591 | 180.1 | 11622 | 1.3 | 11622 | 10 | 11510 | 2.7 | 11370 | 1.9 |
| G4 | 800 | 19176 | 11207 | 1217.8 | 11280 | 4.8 | 10805 | 216.0 | 10945 | 580.5 | 11430 | 11.2 | 11522 | 26.8 | 11596 | 180.1 | 11646 | 5.2 | 11646 | 7 | 11416 | 2.6 | 11459 | 2.1 |
| G5 | 800 | 19176 | 11256 | 1261.8 | 11156 | 3.7 | 10417 | 214.1 | 10869 | 598.2 | 11406 | 11.0 | 11485 | 24.2 | 11575 | 180.1 | 11631 | 1.0 | 11631 | 7 | 11505 | 2.5 | 11408 | 1.7 |
| G6 | 800 | 19176 | 1776 | 1261.6 | 1755 | 6.9 | 1748 | 214.4 | 1435 | 581.2 | 1991 | 11.4 | 2095 | 23.3 | 2130 | 180.1 | 2178 | 3.0 | 2178 | 14 | 1994 | 2.6 | 1907 | 1.7 |
| G7 | 800 | 19176 | 1694 | 1336.4 | 1635 | 5.9 | 1524 | 215.4 | 1273 | 587.5 | 1780 | 11.1 | 1957 | 25.5 | 1972 | 180.1 | 2006 | 3.0 | 2006 | 7 | 1802 | 2.6 | 1804 | 1.8 |
| G8 | 800 | 19176 | 1693 | 1235.2 | 1651 | 6.1 | 1566 | 215.4 | 1241 | 591.8 | 1758 | 11.1 | 1955 | 25.5 | 1974 | 180.1 | 2005 | 5.7 | 2005 | 10 | 1876 | 2.8 | 1775 | 1.8 |
| G9 | 800 | 19176 | 1676 | 1215.0 | 1720 | 8.0 | 1545 | 211.7 | 1345 | 582.3 | 1845 | 14.6 | 2044 | 26.8 | 2006 | 180.1 | 2054 | 3.2 | 2054 | 13 | 1839 | 2.6 | 1876 | 1.9 |
| G10 | 800 | 19176 | 1675 | 1227.3 | 1700 | 7.3 | 1445 | 212.7 | 1313 | 589.5 | 1816 | 10.9 | 1930 | 27.1 | 1953 | 180.1 | 2000 | 68.1 | 2000 | 10 | 1811 | 2.6 | 1755 | 1.8 |
| G11 | 800 | 1600 | N/A | N/A | 466 | 3.0 | 464 | 216.2 | 406 | 509.4 | 540 | 11.0 | 545 | 25.6 | 548 | 180.1 | 564 | 0.2 | 564 | 11 | 496 | 1.8 | 494 | 1.5 |
| G12 | 800 | 1600 | N/A | N/A | 466 | 2.4 | 470 | 215.0 | 388 | 514.8 | 534 | 11.0 | 541 | 27.2 | 542 | 180.1 | 556 | 3.5 | 556 | 16 | 498 | 1.9 | 494 | 1.4 |
| G13 | 800 | 1600 | N/A | N/A | 486 | 3.0 | 480 | 214.2 | 426 | 520.0 | 560 | 10.8 | 565 | 26.9 | 568 | 180.1 | 582 | 0.9 | 582 | 23 | 518 | 1.9 | 524 | 1.5 |
| G14 | 800 | 4694 | 2942 | 1716.6 | 2930 | 3.1 | 2484 | 211.5 | 2855 | 564.2 | 2985 | 11.1 | 2807 | 23.4 | 3036 | 180.1 | 3064 | 251.3 | 3064 | 119 | 2932 | 1.5 | 2953 | 1.8 |
| G15 | 800 | 4661 | N/A | N/A | 2932 | 3.1 | 2416 | 213.0 | 2836 | 547.7 | 2966 | 11.1 | 2741 | 26.2 | 3014 | 180.1 | 3050 | 52.2 | 3050 | 80 | 2920 | 1.8 | 2871 | 1.4 |
| G16 | 800 | 4672 | N/A | N/A | 2937 | 3.8 | 2604 | 212.9 | 2848 | 541.3 | 2987 | 14.3 | 2757 | 26.8 | 3018 | 180.1 | 3052 | 93.7 | 3052 | 69 | 2917 | 1.7 | 2916 | 1.3 |
| G17 | 800 | 4667 | 2916 | 1738.2 | 2922 | 3.3 | 2456 | 186.7 | 2829 | 558.9 | 2967 | 12.1 | 2754 | 25.8 | 3027 | 180.1 | 3047 | 129.5 | 3047 | 104 | 2932 | 1.9 | 2914 | 1.5 |
| G18 | 800 | 4694 | 838 | 871.7 | 825 | 3.7 | 763 | 212.9 | 643 | 567.0 | 922 | 11.2 | 925 | 25.6 | 966 | 180.1 | 992 | 112.7 | 992 | 40 | 903 | 2.1 | 905 | 1.7 |
| G19 | 800 | 4661 | 763 | 1245.4 | 740 | 3.6 | 725 | 206.5 | 571 | 571.2 | 816 | 11.4 | 828 | 27.2 | 881 | 180.1 | 906 | 266.9 | 906 | 49 | 808 | 2 | 772 | 1.5 |
| G20 | 800 | 4672 | 781 | 1015.6 | 767 | 3.5 | 740 | 213.4 | 633 | 565.8 | 860 | 11.9 | 897 | 28.6 | 925 | 180.1 | 941 | 43.7 | 941 | 31 | 843 | 2.1 | 788 | 1.8 |
| G21 | 800 | 4667 | 821 | 1350.3 | 784 | 3.0 | 740 | 209.3 | 620 | 572.2 | 837 | 14.1 | 864 | 25.2 | 925 | 180.1 | 931 | 155.3 | 931 | 32 | 858 | 2.1 | 848 | 1.6 |
| G22 | 2000 | 19990 | N/A | N/A | 12777 | 12.2 | 12283 | 212.9 | N/A | N/A | 13004 | 95.6 | 13169 | 198.4 | 13280 | 180.1 | 13359 | 352.4 | 13359 | 413 | 13028 | 2.6 | 13007 | 2.7 |
| G23 | 2000 | 19990 | N/A | N/A | 12688 | 10.2 | 12314 | 211.7 | N/A | N/A | 12958 | 95.6 | 13096 | 196.7 | 13297 | 180.1 | 13344 | 433.8 | 13342 | 150 | 13048 | 2.9 | 12936 | 1.9 |
| G24 | 2000 | 19990 | N/A | N/A | 12721 | 10.0 | 11606 | 214.5 | N/A | N/A | 13002 | 95.0 | 13096 | 349.2 | 13284 | 180.1 | 13337 | 777.9 | 13337 | 234 | 13035 | 1.9 | 12933 | 2.4 |
| G25 | 2000 | 19990 | N/A | N/A | 12725 | 11.7 | 12233 | 214.3 | N/A | N/A | 12968 | 102.6 | 13146 | 202.6 | 13279 | 180.1 | 13340 | 442.5 | 13328 | 258 | 13040 | 2 | 12947 | 1.9 |
| G26 | 2000 | 19990 | N/A | N/A | 12725 | 10.8 | 12141 | 217.2 | N/A | N/A | 12966 | 96.9 | 13126 | 201.4 | 13253 | 180.1 | 13328 | 535.1 | 13328 | 291 | 13054 | 2.5 | 12954 | 3.5 |
| G27 | 2000 | 19990 | N/A | N/A | 2632 | 11.2 | 2509 | 216.3 | N/A | N/A | 3062 | 98.9 | 3212 | 200.2 | 3300 | 180.1 | 3341 | 42.3 | 3341 | 152 | 2993 | 2.8 | 2971 | 2.1 |
| G28 | 2000 | 19990 | N/A | N/A | 2762 | 11.2 | 2563 | 214.9 | N/A | N/A | 2963 | 96.8 | 3160 | 201.1 | 3265 | 180.1 | 3298 | 707.2 | 3298 | 197 | 2985 | 2.6 | 2923 | 1.9 |
| G29 | 2000 | 19990 | N/A | N/A | 2736 | 12.3 | 2578 | 216.5 | N/A | N/A | 3044 | 96.4 | 3312 | 204.3 | 3348 | 180.1 | 3405 | 555.2 | 3405 | 293 | 3056 | 2.9 | 3089 | 1.9 |
| G30 | 2000 | 19990 | N/A | N/A | 2774 | 11.7 | 2559 | 214.3 | N/A | N/A | 3074 | 99.3 | 3287 | 200.0 | 3363 | 180.1 | 3413 | 330.5 | 3413 | 410 | 3004 | 2.8 | 3025 | 2.9 |
| G31 | 2000 | 19990 | N/A | N/A | 2736 | 11.5 | 2539 | 216.3 | N/A | N/A | 2998 | 96.3 | 3215 | 201.4 | 3241 | 180.1 | 3310 | 592.6 | 3310 | 412 | 3015 | 2.1 | 2943 | 1.9 |
| G32 | 2000 | 4000 | N/A | N/A | 1136 | 6.8 | 1106 | 214.9 | N/A | N/A | 1338 | 92.7 | 1349 | 198.7 | 1360 | 180.1 | 1410 | 65.8 | 1410 | 330 | 1240 | 2.2 | 1226 | 1.7 |
| G33 | 2000 | 4000 | N/A | N/A | 1106 | 6.6 | 1068 | 213.4 | N/A | N/A | 1302 | 89.3 | 1330 | 194.3 | 1342 | 180.1 | 1382 | 504.1 | 1382 | 349 | 1224 | 2.3 | 1208 | 1.7 |
| G34 | 2000 | 4000 | N/A | N/A | 1118 | 5.8 | 1106 | 212.4 | N/A | N/A | 1314 | 95.6 | 1320 | 198.7 | 1350 | 180.1 | 1384 | 84.2 | 1384 | 302 | 1238 | 2.3 | 1220 | 1.6 |
| G35 | 2000 | 11778 | N/A | N/A | 7358 | 9.4 | 6196 | 185.7 | N/A | N/A | 7495 | 95.2 | 6599 | 200.6 | 7624 | 180.1 | 7686 | 796.7 | 7686 | 1070 | 7245 | 1.9 | 7260 | 1.9 |

Table 6: Continued.

| Instance | $|\mathcal{V}|$ | $|\mathcal{E}|$ | GW Obj.↑ | GW Time (s)↓ | MD Obj.↑ | MD Time (s)↓ | PT-GNN Obj.↑ | PT-GNN Time (s)↓ | Genetic Obj.↑ | Genetic Time (s)↓ | BQP Obj.↑ | BQP Time (s)↓ | ECO-DQN Obj.↑ | ECO-DQN Time (s)↓ | ANYCSP Obj.↑ | ANYCSP Time (s)↓ | MOH Obj.↑ | MOH Time (s)↓ | LPI Obj.↑ | LPI Time (s)↓ | ROS-vanilla Obj.↑ | ROS-vanilla Time (s)↓ | ROS Obj.↑ | ROS Time (s)↓ |
|---|---|---|---|---|---|---|---|---|---|---|---|---|---|---|---|---|---|---|---|---|---|---|---|---|
| G36 | 2000 | 11766 | N/A | N/A | 7336 | 10.1 | 6424 | 214.8 | N/A | N/A | 7490 | 95.3 | 6602 | 195.1 | 7628 | 180.1 | 7680 | 664.5 | 7680 | 5790 | 7235 | 2.4 | 7107 | 1.5 |
| G37 | 2000 | 11785 | N/A | N/A | 7400 | 9.3 | 6224 | 185.3 | N/A | N/A | 7498 | 95.4 | 6555 | 204.1 | 7617 | 180.1 | 7691 | 652.8 | 7691 | 4082 | 7164 | 1.7 | 7141 | 1.5 |
| G38 | 2000 | 11779 | N/A | N/A | 7343 | 8.6 | 6841 | 212.8 | N/A | N/A | 7507 | 100.6 | 6655 | 200.3 | 7629 | 180.1 | 7688 | 779.7 | 7688 | 614 | 7114 | 1.6 | 7173 | 1.8 |
| G39 | 2000 | 11778 | N/A | N/A | 1998 | 9.2 | 1853 | 215.5 | N/A | N/A | 2196 | 94.4 | 1904 | 198.8 | 2354 | 180.1 | 2408 | 787.7 | 2408 | 347 | 2107 | 2.5 | 2165 | 1.7 |
| G40 | 2000 | 11766 | N/A | N/A | 1971 | 9.0 | 1855 | 216.0 | N/A | N/A | 2169 | 97.3 | 2171 | 199.8 | 2320 | 180.1 | 2400 | 472.5 | 2400 | 314 | 2207 | 2.7 | 2128 | 2.5 |
| G41 | 2000 | 11785 | N/A | N/A | 1969 | 9.1 | 1898 | 218.6 | N/A | N/A | 2183 | 105.8 | 1925 | 203.7 | 2346 | 180.1 | 2405 | 377.4 | 2405 | 286 | 2120 | 1.6 | 2139 | 2.2 |
| G42 | 2000 | 11779 | N/A | N/A | 2075 | 9.5 | 1933 | 206.3 | N/A | N/A | 2255 | 95.5 | 2152 | 200.1 | 2416 | 180.1 | 2481 | 777.4 | 2481 | 328 | 2200 | 2.2 | 2235 | 2.4 |
| G43 | 1000 | 9990 | 6340 | 1784.5 | 6380 | 5.0 | 6049 | 186.9 | 5976 | 914.4 | 6509 | 18.0 | 6585 | 44.3 | 6631 | 180.1 | 6660 | 1.2 | 6660 | 19 | 6539 | 2.7 | 6471 | 1.7 |
| G44 | 1000 | 9990 | 6351 | 1486.7 | 6327 | 5.0 | 6099 | 190.8 | 6009 | 914.3 | 6463 | 18.5 | 6577 | 42.2 | 6632 | 180.1 | 6650 | 5.3 | 6650 | 20 | 6498 | 2.5 | 6472 | 1.7 |
| G45 | 1000 | 9990 | 6355 | 1582.0 | 6329 | 4.9 | 6091 | 189.6 | 6006 | 921.5 | 6489 | 22.4 | 6581 | 41.3 | 6632 | 180.1 | 6654 | 6.9 | 6654 | 19 | 6528 | 2.4 | 6489 | 1.7 |
| G46 | 1000 | 9990 | 6357 | 1612.8 | 6300 | 4.8 | 5594 | 188.6 | 5978 | 916.2 | 6485 | 18.4 | 6570 | 43.8 | 6631 | 180.1 | 6649 | 67.3 | 6649 | 21 | 6498 | 2.5 | 6499 | 2.5 |
| G47 | 1000 | 9990 | N/A | N/A | 6369 | 4.7 | 6049 | 184.8 | 5948 | 912.4 | 6491 | 18.4 | 6575 | 46.8 | 6655 | 180.1 | 6657 | 43.3 | 6657 | 25 | 6497 | 2.5 | 6489 | 1.8 |
| G48 | 3000 | 6000 | N/A | N/A | 5006 | 10.6 | 4958 | 191.4 | N/A | N/A | 6000 | 300.4 | 5879 | 881.2 | 5962 | 180.2 | 6000 | 0.0 | 6000 | 94 | 5640 | 3.2 | 5498 | 2.1 |
| G49 | 3000 | 6000 | N/A | N/A | 5086 | 10.1 | 4938 | 192.0 | N/A | N/A | 6000 | 303.0 | 5879 | 871.5 | 5936 | 180.2 | 6000 | 0.0 | 6000 | 93 | 5580 | 3.1 | 5452 | 2.2 |
| G50 | 3000 | 6000 | N/A | N/A | 5156 | 11.3 | 4948 | 192.3 | N/A | N/A | 5880 | 299.8 | 5807 | 876.7 | 5830 | 180.2 | 5880 | 532.1 | 5880 | 90 | 5656 | 3.2 | 5582 | 1.9 |
| G51 | 1000 | 5909 | N/A | N/A | 3693 | 4.1 | 3293 | 188.5 | 3568 | 887.9 | 3759 | 17.7 | 3413 | 45.0 | 3821 | 180.2 | 3848 | 189.2 | 3848 | 145 | 3629 | 1.5 | 3677 | 1.7 |
| G52 | 1000 | 5916 | N/A | N/A | 3695 | 4.7 | 3185 | 183.8 | 3575 | 897.7 | 3771 | 18.5 | 3441 | 41.9 | 3836 | 180.2 | 3851 | 209.7 | 3851 | 119 | 3526 | 1.3 | 3641 | 1.6 |
| G53 | 1000 | 5914 | N/A | N/A | 3670 | 4.5 | 3029 | 170.6 | 3545 | 872.8 | 3752 | 18.0 | 3469 | 38.3 | 3807 | 180.2 | 3850 | 299.3 | 3850 | 182 | 3633 | 1.5 | 3658 | 1.6 |
| G54 | 1000 | 5916 | N/A | N/A | 3682 | 4.4 | 3201 | 189.3 | 3548 | 880.1 | 3753 | 18.0 | 3485 | 44.6 | 3820 | 180.2 | 3852 | 190.4 | 3852 | 140 | 3653 | 1.6 | 3642 | 1.3 |
| G55 | 5000 | 12498 | N/A | N/A | 9462 | 24.4 | 9110 | 201.5 | N/A | N/A | 9862 | 1142.1 | N/A | N/A | 10213 | 180.2 | 10299 | 1230.4 | 10299 | 6594 | 9819 | 2.1 | 9779 | 2.9 |
| G56 | 5000 | 12498 | N/A | N/A | 3203 | 23.8 | 2939 | 201.8 | N/A | N/A | 3710 | 1147.6 | N/A | N/A | 3918 | 180.2 | 4016 | 990.4 | 4017 | 49445 | 3444 | 2 | 3475 | 2.5 |
| G57 | 5000 | 10000 | N/A | N/A | 2770 | 17.3 | 2650 | 203.0 | N/A | N/A | 3310 | 1120.8 | N/A | N/A | 3404 | 180.2 | 3494 | 1528.3 | 3494 | 3494 | 3040 | 1.7 | 3078 | 2.5 |
| G58 | 5000 | 29570 | N/A | N/A | 18452 | 29.2 | 17115 | 202.8 | N/A | N/A | 18813 | 1176.6 | N/A | N/A | 19152 | 180.2 | 19288 | 1522.3 | 19294 | 65737 | 17632 | 2.3 | 17574 | 1.8 |
| G59 | 5000 | 29570 | N/A | N/A | 5099 | 31.6 | 4674 | 202.6 | N/A | N/A | 5490 | 1183.4 | N/A | N/A | 5952 | 180.2 | 6087 | 2498.8 | 6088 | 65112 | 5343 | 1.9 | 5407 | 4.7 |
| G60 | 7000 | 17148 | N/A | N/A | 13004 | 34.8 | 11430 | 214.0 | N/A | N/A | N/A | N/A | N/A | N/A | 14096 | 180.2 | 14190 | 2945.4 | 14190 | 44802 | 13433 | 2 | 13402 | 2 |
| G61 | 7000 | 17148 | N/A | N/A | 4592 | 36.0 | 4225 | 213.9 | N/A | N/A | N/A | N/A | N/A | N/A | 5710 | 180.2 | 5798 | 6603.3 | 5798 | 74373 | 5037 | 3.8 | 5011 | 2 |
| G62 | 7000 | 14000 | N/A | N/A | 3922 | 26.1 | 3720 | 214.0 | N/A | N/A | N/A | N/A | N/A | N/A | 4732 | 180.2 | 4868 | 5568.6 | 4872 | 26537 | 4252 | 3.8 | 4294 | 2.8 |
| G63 | 7000 | 41459 | N/A | N/A | 25938 | 45.1 | 22224 | 213.9 | N/A | N/A | N/A | N/A | N/A | N/A | 26884 | 180.2 | 27033 | 6492.1 | 27033 | 52726 | 24185 | 1.7 | 24270 | 1.5 |
| G64 | 7000 | 41459 | N/A | N/A | 7283 | 43.7 | 6616 | 212.9 | N/A | N/A | N/A | N/A | N/A | N/A | 8514 | 180.4 | 8747 | 4011.1 | 8752 | 49158 | 7508 | 2.3 | 7657 | 3 |
| G65 | 8000 | 16000 | N/A | N/A | 4520 | 32.5 | 4208 | 220.4 | N/A | N/A | N/A | N/A | N/A | N/A | 5392 | 180.1 | 5560 | 4709.5 | 5562 | 21737 | 4878 | 4.4 | 4826 | 2.5 |
| G66 | 9000 | 18000 | N/A | N/A | 5100 | 37.3 | 4816 | 223.1 | N/A | N/A | N/A | N/A | N/A | N/A | 6172 | 159.6 | 6360 | 6061.9 | 6364 | 34062 | 5570 | 5.5 | 5580 | 3.3 |
| G67 | 10000 | 20000 | N/A | N/A | 5592 | 43.4 | 5312 | 258.9 | N/A | N/A | N/A | N/A | N/A | N/A | 6772 | 146.5 | 6942 | 4214.3 | 6948 | 61556 | 6090 | 6.2 | 6010 | 1.9 |
| G70 | 10000 | 9999 | N/A | N/A | 8551 | 54.3 | 8404 | 217.5 | N/A | N/A | N/A | N/A | N/A | N/A | 9300 | 180.2 | 9544 | 8732.4 | 9594 | 28820 | 9004 | 4.9 | 8916 | 3.4 |
| G72 | 10000 | 20000 | N/A | N/A | 5638 | 44.2 | 5386 | 217.6 | N/A | N/A | N/A | N/A | N/A | N/A | 6826 | 180.2 | 6998 | 6586.6 | 7004 | 42542 | 6066 | 6.2 | 6102 | 3.9 |
| G77 | 14000 | 28000 | N/A | N/A | 7934 | 66.0 | 7352 | 291.0 | N/A | N/A | N/A | N/A | N/A | N/A | 9694 | 180.2 | 9928 | 9863.6 | 9926 | 66662 | 8678 | 9 | 8740 | 8.1 |
| G81 | 20000 | 40000 | N/A | N/A | 11226 | 130.8 | 10582 | 494.1 | N/A | N/A | N/A | N/A | N/A | N/A | 13684 | 180.2 | 14036 | 20422.0 | 14030 | 66691 | 12260 | 13.7 | 12332 | 9.3 |

Table 7: Complete results on Gset instances for Max-3-Cut.

| Instance | $|\mathcal{V}|$ | $|\mathcal{E}|$ | MD Obj. ↑ | MD Time (s) ↓ | Genetic Obj. ↑ | Genetic Time (s) ↓ | BQP Obj. ↑ | BQP Time (s) ↓ | ANYCSP Obj. ↑ | ANYCSP Time (s) ↓ | MOH Obj. ↑ | MOH Time (s) ↓ | ROS-vanilla Obj. ↑ | ROS-vanilla Time (s) ↓ | ROS Obj. ↑ | ROS Time (s) ↓ |
|---|---|---|---|---|---|---|---|---|---|---|---|---|---|---|---|---|
| G1 | 800 | 19176 | 14735 | 9.6 | 14075 | 595.3 | 14880 | 16.5 | 15115 | 180.1 | 15165 | 557.3 | 14949 | 2.8 | 14961 | 1.9 |
| G2 | 800 | 19176 | 14787 | 8.4 | 14035 | 595.3 | 14845 | 17.0 | 15088 | 180.1 | 15172 | 333.3 | 15033 | 2.8 | 14932 | 2.3 |
| G3 | 800 | 19176 | 14663 | 6.5 | 14105 | 588.6 | 14872 | 17.0 | 15111 | 180.1 | 15173 | 269.6 | 15016 | 2.9 | 14914 | 1.9 |
| G4 | 800 | 19176 | 14716 | 6.9 | 14055 | 588.7 | 14886 | 17.1 | 15115 | 180.1 | 15184 | 300.6 | 14984 | 3.3 | 14961 | 1.9 |
| G5 | 800 | 19176 | 14681 | 8.1 | 14104 | 591.9 | 14847 | 17.3 | 15092 | 180.1 | 15193 | 98.2 | 15006 | 3.2 | 14962 | 2.9 |
| G6 | 800 | 19176 | 2161 | 7.8 | 1504 | 604.4 | 2302 | 25.0 | 1164 | 180.1 | 2632 | 307.3 | 2436 | 2.8 | 2361 | 1.8 |
| G7 | 800 | 19176 | 2017 | 8.9 | 1260 | 589.9 | 2081 | 16.6 | 932 | 180.1 | 2409 | 381.0 | 2188 | 2.1 | 2188 | 2.4 |
| G8 | 800 | 19176 | 1938 | 7.7 | 1252 | 589.7 | 2096 | 19.3 | 1007 | 180.1 | 2428 | 456.5 | 2237 | 2.8 | 2171 | 2.1 |
| G9 | 800 | 19176 | 2031 | 8.2 | 1326 | 604.4 | 2099 | 16.5 | 1164 | 180.1 | 2478 | 282.0 | 2246 | 2.8 | 2185 | 2.2 |
| G10 | 800 | 19176 | 1961 | 7.5 | 1266 | 593.3 | 2055 | 18.2 | 919 | 180.1 | 2407 | 569.3 | 2201 | 2.9 | 2181 | 2.3 |
| G11 | 800 | 1600 | 553 | 4.0 | 414 | 554.5 | 624 | 16.4 | 650 | 180.1 | 669 | 143.8 | 616 | 2 | 591 | 1.4 |
| G12 | 800 | 1600 | 530 | 4.4 | 388 | 543.6 | 608 | 17.4 | 633 | 180.1 | 660 | 100.7 | 604 | 2 | 582 | 1.5 |
| G13 | 800 | 1600 | 558 | 4.0 | 425 | 550.8 | 638 | 18.9 | 663 | 180.1 | 686 | 459.4 | 617 | 2 | 629 | 1.4 |
| G14 | 800 | 4694 | 3844 | 5.0 | 3679 | 571.1 | 3900 | 16.9 | 3973 | 180.1 | 4012 | 88.2 | 3914 | 2.8 | 3892 | 2.1 |
| G15 | 800 | 4661 | 3815 | 4.8 | 3625 | 567.6 | 3885 | 17.3 | 3975 | 180.1 | 3984 | 80.3 | 3817 | 1.9 | 3838 | 2 |
| G16 | 800 | 4672 | 3825 | 5.3 | 3642 | 561.5 | 3896 | 18.2 | 3945 | 180.1 | 3991 | 1.3 | 3843 | 2.3 | 3845 | 1.6 |
| G17 | 800 | 4667 | 3815 | 5.3 | 3640 | 558.7 | 3886 | 20.2 | 3955 | 180.1 | 3983 | 7.8 | 3841 | 2.4 | 3852 | 1.6 |
| G18 | 800 | 4694 | 992 | 4.5 | 704 | 584.0 | 1083 | 18.7 | 999 | 180.1 | 1207 | 0.3 | 1094 | 2.2 | 1067 | 1.7 |
| G19 | 800 | 4661 | 869 | 4.4 | 595 | 584.2 | 962 | 17.0 | 915 | 180.1 | 1081 | 0.2 | 972 | 2.1 | 967 | 1.7 |
| G20 | 800 | 4672 | 928 | 4.5 | 589 | 576.8 | 977 | 17.0 | 861 | 180.1 | 1122 | 13.3 | 1006 | 2.2 | 993 | 1.8 |
| G21 | 800 | 4667 | 936 | 4.9 | 612 | 576.3 | 984 | 17.5 | 895 | 180.1 | 1109 | 55.8 | 1011 | 2.2 | 975 | 1.5 |
| G22 | 2000 | 19990 | 16402 | 15.2 | N/A | N/A | 16599 | 135.5 | 17098 | 180.1 | 17167 | 28.5 | 16790 | 3.3 | 16601 | 2.2 |
| G23 | 2000 | 19990 | 16422 | 15.0 | N/A | N/A | 16626 | 135.6 | 17049 | 180.1 | 17168 | 45.1 | 16819 | 3.9 | 16702 | 2.1 |
| G24 | 2000 | 19990 | 16452 | 16.1 | N/A | N/A | 16591 | 137.7 | 17042 | 180.1 | 17162 | 16.3 | 16801 | 3.6 | 16754 | 3 |
| G25 | 2000 | 19990 | 16407 | 16.2 | N/A | N/A | 16661 | 141.8 | 17085 | 180.1 | 17163 | 64.8 | 16795 | 2.1 | 16673 | 1.8 |
| G26 | 2000 | 19990 | 16422 | 15.3 | N/A | N/A | 16608 | 136.3 | 17014 | 180.1 | 17154 | 44.8 | 16758 | 3.1 | 16665 | 2 |
| G27 | 2000 | 19990 | 3250 | 16.4 | N/A | N/A | 3475 | 134.3 | 2846 | 180.1 | 4020 | 53.2 | 3517 | 1.7 | 3532 | 2 |
| G28 | 2000 | 19990 | 3198 | 16.1 | N/A | N/A | 3433 | 136.4 | 2778 | 180.1 | 3973 | 38.9 | 3507 | 3 | 3414 | 2.1 |
| G29 | 2000 | 19990 | 3324 | 16.0 | N/A | N/A | 3582 | 136.2 | 3035 | 180.1 | 4106 | 68.2 | 3634 | 3.4 | 3596 | 2 |
| G30 | 2000 | 19990 | 3320 | 16.2 | N/A | N/A | 3578 | 133.6 | 3032 | 180.1 | 4119 | 150.4 | 3656 | 3.1 | 3654 | 3.4 |
| G31 | 2000 | 19990 | 3243 | 17.0 | N/A | N/A | 3439 | 131.0 | 2881 | 180.1 | 4003 | 124.7 | 3596 | 3 | 3525 | 2.5 |
| G32 | 2000 | 4000 | 1342 | 11.1 | N/A | N/A | 1545 | 129.3 | 1590 | 180.1 | 1653 | 160.1 | 1488 | 2.5 | 1482 | 1.7 |
| G33 | 2000 | 4000 | 1284 | 10.7 | N/A | N/A | 1517 | 126.2 | 1550 | 180.1 | 1625 | 62.6 | 1449 | 2.5 | 1454 | 2 |
| G34 | 2000 | 4000 | 1292 | 10.9 | N/A | N/A | 1499 | 126.0 | 1525 | 180.1 | 1607 | 88.9 | 1418 | 2.4 | 1435 | 1.7 |
| G35 | 2000 | 11778 | 9644 | 14.2 | N/A | N/A | 9816 | 138.1 | 9968 | 180.1 | 10046 | 66.2 | 9225 | 2 | 9536 | 1.7 |

Table 7: Continued.

| Instance | $\|\mathcal{V}\|$ | $\|\mathcal{E}\|$ | MD Obj. ↑ | MD Time (s) ↓ | Genetic Obj. ↑ | Genetic Time (s) ↓ | BQP Obj. ↑ | BQP Time (s) ↓ | ANYCSP Obj. ↑ | ANYCSP Time (s) ↓ | MOH Obj. ↑ | MOH Time (s) ↓ | ROS-vanilla Obj. ↑ | ROS-vanilla Time (s) ↓ | ROS Obj. | ROS Time |
|---|---|---|---|---|---|---|---|---|---|---|---|---|---|---|---|---|
| G36 | 2000 | 11766 | 9600 | 13.6 | N/A | N/A | 9786 | 138.6 | 9972 | 180.1 | 10039 | 74.3 | 9372 | 2.1 | 9581 | 2.3 |
| G37 | 2000 | 11785 | 9632 | 14.9 | N/A | N/A | 9821 | 139.2 | 9983 | 180.1 | 10052 | 3.4 | 8893 | 1.4 | 9422 | 1.5 |
| G38 | 2000 | 11779 | 9629 | 14.0 | N/A | N/A | 9775 | 142.3 | 9980 | 180.1 | 10040 | 116.6 | 9489 | 2.5 | 9370 | 1.5 |
| G39 | 2000 | 11778 | 2368 | 13.4 | N/A | N/A | 2600 | 132.8 | 2497 | 180.1 | 2903 | 9.0 | 2621 | 2.5 | 2557 | 2.2 |
| G40 | 2000 | 11766 | 2315 | 13.3 | N/A | N/A | 2568 | 131.2 | 2428 | 8.6 | 2870 | 82.8 | 2474 | 2 | 2524 | 2.4 |
| G41 | 2000 | 11785 | 2386 | 12.7 | N/A | N/A | 2606 | 129.9 | 2416 | 8.4 | 2887 | 87.7 | 2521 | 3.2 | 2584 | 2.5 |
| G42 | 2000 | 11779 | 2490 | 13.1 | N/A | N/A | 2682 | 129.2 | 2685 | 32.8 | 2980 | 2.5 | 2638 | 2.7 | 2613 | 2.2 |
| G43 | 1000 | 9990 | 8214 | 8.1 | 7624 | 926.7 | 8329 | 29.9 | 8531 | 180.1 | 8573 | 380.3 | 8414 | 2.6 | 8349 | 2.3 |
| G44 | 1000 | 9990 | 8187 | 7.0 | 7617 | 919.0 | 8326 | 27.7 | 8515 | 180.1 | 8571 | 616.8 | 8369 | 2.6 | 8311 | 1.7 |
| G45 | 1000 | 9990 | 8226 | 7.7 | 7602 | 926.7 | 8296 | 34.2 | 8530 | 180.1 | 8566 | 186.2 | 8397 | 2.9 | 8342 | 1.8 |
| G46 | 1000 | 9990 | 8229 | 7.5 | 7635 | 918.7 | 8312 | 27.8 | 8501 | 180.1 | 8568 | 215.3 | 8409 | 2.6 | 8339 | 1.7 |
| G47 | 1000 | 9990 | 8211 | 7.2 | 7619 | 928.0 | 8322 | 27.3 | 8513 | 180.2 | 8572 | 239.4 | 8386 | 2.6 | 8357 | 2.2 |
| G48 | 3000 | 6000 | 5806 | 14.7 | N/A | N/A | 5998 | 394.8 | 5985 | 180.2 | 6000 | 0.4 | 5954 | 2.8 | 5912 | 2 |
| G49 | 3000 | 6000 | 5794 | 14.4 | N/A | N/A | 5998 | 404.0 | 5974 | 180.2 | 6000 | 0.9 | 5938 | 2.8 | 5914 | 1.8 |
| G50 | 3000 | 6000 | 5823 | 14.5 | N/A | N/A | 6000 | 427.1 | 5989 | 180.2 | 6000 | 119.2 | 5938 | 2.9 | 5918 | 1.8 |
| G51 | 1000 | 5909 | 4805 | 6.6 | 4582 | 889.5 | 4922 | 28.6 | 4990 | 180.2 | 5037 | 47.9 | 4814 | 2.4 | 4820 | 1.7 |
| G52 | 1000 | 5916 | 4849 | 6.4 | 4571 | 908.1 | 4910 | 27.8 | 5002 | 180.2 | 5040 | 0.7 | 4796 | 1.9 | 4866 | 1.9 |
| G53 | 1000 | 5914 | 4845 | 6.8 | 4568 | 898.6 | 4920 | 27.6 | 5005 | 180.2 | 5039 | 223.9 | 4846 | 2.6 | 4808 | 1.6 |
| G54 | 1000 | 5916 | 4836 | 6.4 | 4562 | 911.7 | 4921 | 30.1 | 4998 | 180.2 | 5036 | 134.0 | 4833 | 2.2 | 4785 | 1.4 |
| G55 | 5000 | 12498 | 11612 | 37.9 | N/A | N/A | 12042 | 1506.0 | 12355 | 180.2 | 12429 | 383.1 | 12010 | 2.1 | 11965 | 2.6 |
| G56 | 5000 | 12498 | 3716 | 38.5 | N/A | N/A | 4205 | 1341.5 | 4408 | 180.2 | 4752 | 569.2 | 4085 | 3.3 | 4037 | 2.1 |
| G57 | 5000 | 10000 | 3246 | 33.0 | N/A | N/A | 3817 | 1317.2 | 3913 | 180.2 | 4083 | 535.6 | 3597 | 3.3 | 3595 | 2.8 |
| G58 | 5000 | 29570 | 24099 | 47.1 | N/A | N/A | 24603 | 1468.3 | 25025 | 180.2 | 25195 | 576.0 | 22748 | 2.1 | 23274 | 1.9 |
| G59 | 5000 | 29570 | 6057 | 46.3 | N/A | N/A | 6631 | 1377.1 | 6178 | 180.2 | 7262 | 27.5 | 6133 | 1.7 | 6448 | 3.5 |
| G60 | 7000 | 17148 | 15993 | 58.5 | N/A | N/A | N/A | N/A | 16974 | 180.2 | 17076 | 683.0 | 16467 | 2.6 | 16398 | 2.3 |
| G61 | 7000 | 17148 | 5374 | 57.7 | N/A | N/A | N/A | N/A | 6426 | 180.2 | 6853 | 503.1 | 5881 | 2.5 | 5861 | 3.6 |
| G62 | 7000 | 14000 | 4497 | 49.7 | N/A | N/A | N/A | N/A | 5444 | 180.2 | 5685 | 242.4 | 4983 | 3.4 | 5086 | 2.7 |
| G63 | 7000 | 41459 | 33861 | 73.4 | N/A | N/A | N/A | N/A | 35070 | 180.2 | 35322 | 658.5 | 32868 | 4 | 31926 | 1.9 |
| G64 | 7000 | 41459 | 8773 | 73.4 | N/A | N/A | N/A | N/A | 8557 | 180.4 | 10443 | 186.9 | 8911 | 2.8 | 9171 | 2.5 |
| G65 | 8000 | 16000 | 5212 | 59.6 | N/A | N/A | N/A | N/A | 6232 | 180.1 | 6490 | 324.7 | 5735 | 3.5 | 5775 | 2.6 |
| G66 | 9000 | 18000 | 5948 | 69.0 | N/A | N/A | N/A | N/A | 7129 | 159.6 | 7416 | 542.5 | 6501 | 5.4 | 6610 | 3.9 |
| G67 | 10000 | 20000 | 6545 | 79.0 | N/A | N/A | N/A | N/A | 7827 | 146.5 | 8086 | 756.7 | 7001 | 3.5 | 7259 | 4.1 |
| G70 | 10000 | 9999 | 9718 | 74.8 | N/A | N/A | N/A | N/A | 9848 | 180.2 | 9999 | 7.8 | 9982 | 4.2 | 9971 | 2.5 |
| G72 | 10000 | 20000 | 6612 | 79.2 | N/A | N/A | N/A | N/A | 7893 | 180.2 | 8192 | 271.2 | 7210 | 5.1 | 7297 | 3.5 |
| G77 | 14000 | 28000 | 9294 | 142.3 | N/A | N/A | N/A | N/A | 11128 | 180.2 | 11578 | 154.9 | 10191 | 8.6 | 10329 | 8.5 |
| G81 | 20000 | 40000 | 13098 | 241.1 | N/A | N/A | N/A | N/A | 15658 | 180.2 | 16321 | 331.2 | 14418 | 20.2 | 14464 | 9.7 |

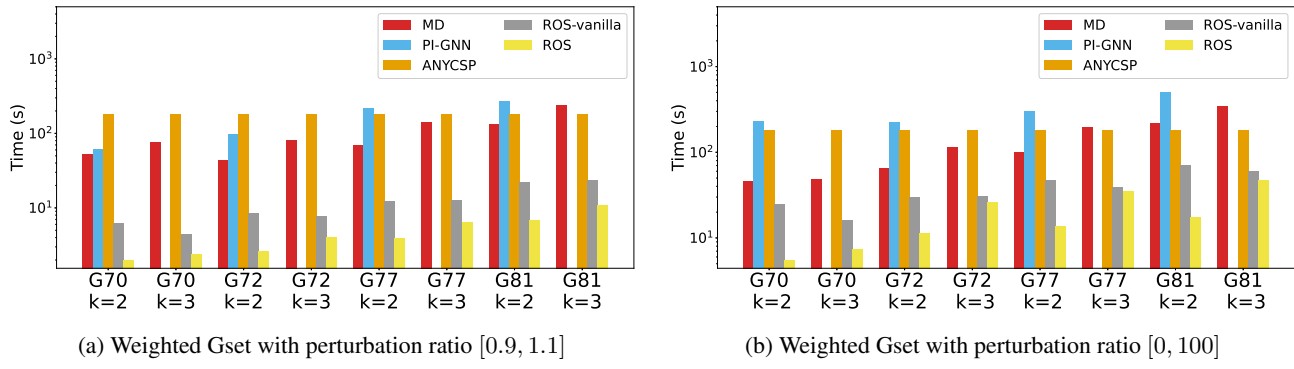

(a) Weighted Gset with perturbation ratio $[0.9, 1.1]$       (b) Weighted Gset with perturbation ratio $[0, 100]$

Figure 4: The computational time comparison of Max-$k$-Cut problems.

Table 8: Cut value comparison of Max-$k$-Cut problems on weighted Gset instances with perturbation ratio $[0.9, 1.1]$.

| Methods | G70 (N=10,000) | | G72 (N=10,000) | | G77 (N=14,000) | | G81 (N=20,000) | |
|---|---|---|---|---|---|---|---|---|
| | $k=2$ | $k=3$ | $k=2$ | $k=3$ | $k=2$ | $k=3$ | $k=2$ | $k=3$ |
| MD | 8594.38 | 9709.56 | 5647.28 | 6585.04 | 8051.81 | 9337.31 | 11326.30 | 13179.33 |
| PI-GNN | 8422.79 | – | 5309.65 | – | 7470.89 | – | 10416.44 | – |
| ANYCSP | 5198.87 | 5375.92 | $-15.57$ | $-25.33$ | 81.76 | 114.36 | 33.49 | $-4.25$ |
| ROS-vanilla | 9177.21 | 9991.95 | 6542.78 | 7733.87 | 9265.65 | 10944.35 | 13132.52 | 15456.28 |
| ROS | 8941.80 | 9970.72 | 6165.62 | 7366.54 | 8737.59 | 10359.25 | 12325.85 | 14570.04 |

## E. The Results on Weighted Gset Instances

The computational time on weighted Gset with perturbation ratio of $[0.9, 1.1]$ and $[0, 100]$ are shown in Fig. 4a and Fig. 4b respectively. The cut values are shown in Table 8 and Table 9 respectively.

## F. Ablation Study

### F.1. Model Ablation

We conducted additional ablation studies to clarify the contributions of different modules.

**Effect of Neural Networks:** We consider two cases: (i) replace GNNs by multi-layer perceptrons (denoted by `ROS-MLP`) in our ROS framework and (ii) solve the relaxation via mirror descent (denoted by `MD`). Experiments on the Gset dataset show that `ROS` consistently outperforms `ROS-MLP` and `MD`, highlighting the benefits of using GNNs for the relaxation step.

**Effect of Random Sampling:** We compared `ROS` with `PI-GNN`, which employs heuristic rounding instead of our random sampling algorithm. Results indicate that `ROS` generally outperforms `PI-GNN`, demonstrating the importance of the sampling procedure.

These comparisons, detailed in Tables 10 and 11, confirm that both the GNN-based optimization and the random sampling algorithm contribute significantly to the overall performance.

### F.2. Sample Effect Ablation

We investigated the effect of the number of sampling iterations and report the results in Tables 12, 13, 14, and 15.

**Cut Value** (Table 12, Table 14): The cut values stabilize after approximately 5 sampling iterations, demonstrating strong performance without requiring extensive sampling.

**Sampling Time** (Table 13, Table 15): The time spent on sampling remains negligible compared to the total computational

Table 9: Cut value comparison of Max-$k$-Cut problems on weighted Gset instances with perturbation ratio $[0, 100]$.

| Methods | G70 (N=10,000) | | G72 (N=10,000) | | G77 (N=14,000) | | G81 (N=20,000) | |
|---|---|---|---|---|---|---|---|---|
| | $k = 2$ | $k = 3$ | $k = 2$ | $k = 3$ | $k = 2$ | $k = 3$ | $k = 2$ | $k = 3$ |
| MD | 456581.30 | 497167.48 | 338533.07 | 392908.80 | 482413.19 | 558264.21 | 682809.47 | 790089.41 |
| PI-GNN | 442650.59 | – | 312802.48 | – | 442354.44 | – | 623256.74 | – |
| ANYCSP | 467696.98 | 491654.75 | −1903.50 | −2498.86 | 9712.13 | 10130.89 | 2842.64 | 2845.46 |
| ROS-vanilla | 472067.14 | 498273.60 | 367795.62 | 421189.97 | 524010.92 | 597597.50 | 742432.41 | 846395.85 |
| ROS | 470268.97 | 498269.90 | 362910.89 | 415905.88 | 515991.31 | 590312.40 | 731468.67 | 835424.19 |

Table 10: Cut values returned by each method on Gset.

| Methods | G70 | | G72 | | G77 | | G81 | |
|---|---|---|---|---|---|---|---|---|
| | $k = 2$ | $k = 3$ | $k = 2$ | $k = 3$ | $k = 2$ | $k = 3$ | $k = 2$ | $k = 3$ |
| ROS-MLP | 8867 | 9943 | 6052 | 6854 | 8287 | 9302 | 12238 | 12298 |
| PI-GNN | 8956 | – | 4544 | – | 6406 | – | 8970 | – |
| MD | 8551 | 9728 | 5638 | 6612 | 7934 | 9294 | 11226 | 13098 |
| ROS | 8916 | 9971 | 6102 | 7297 | 8740 | 10329 | 12332 | 14464 |

time, even with an increased number of samples.

These results highlight the efficiency of our sampling method, achieving stable and robust performance with little computational cost.

Table 11: Computational time for each method on Gset.

| Methods | G70 | | G72 | | G77 | | G81 | |
|---|---|---|---|---|---|---|---|---|
| | $k=2$ | $k=3$ | $k=2$ | $k=3$ | $k=2$ | $k=3$ | $k=2$ | $k=3$ |
| ROS-MLP | 3.49 | 3.71 | 3.93 | 4.06 | 8.39 | 9.29 | 11.98 | 16.97 |
| PI-GNN | 34.50 | – | 253.00 | – | 349.40 | – | 557.70 | – |
| MD | 54.30 | 74.80 | 44.20 | 79.20 | 66.00 | 142.30 | 130.80 | 241.10 |
| ROS | 3.40 | 2.50 | 3.90 | 3.50 | 8.10 | 8.50 | 9.30 | 9.70 |

Table 12: Cut value results corresponding to the times of sample $T$ on Gset.

| $T$ | G70 | | G72 | | G77 | | G81 | |
|---|---|---|---|---|---|---|---|---|
| | $k=2$ | $k=3$ | $k=2$ | $k=3$ | $k=2$ | $k=3$ | $k=2$ | $k=3$ |
| 0 | 8912.62 | 9968.11 | 6099.88 | 7304.45 | 8736.58 | 10323.61 | 12328.83 | 14458.09 |
| 1 | 8911 | 9968 | 6100 | 7305 | 8736 | 10321 | 12328 | 14460 |
| 5 | 8915 | 9969 | 6102 | 7304 | 8740 | 10326 | 12332 | 14462 |
| 10 | 8915 | 9971 | 6102 | 7305 | 8740 | 10324 | 12332 | 14459 |
| 25 | 8915 | 9971 | 6102 | 7307 | 8740 | 10326 | 12332 | 14460 |
| 50 | 8915 | 9971 | 6102 | 7307 | 8740 | 10327 | 12332 | 14461 |
| 100 | 8916 | 9971 | 6102 | 7308 | 8740 | 10327 | 12332 | 14462 |

Table 13: Sampling time results corresponding to the times of sample $T$ on Gset.

| $T$ | G70 | | G72 | | G77 | | G81 | |
|---|---|---|---|---|---|---|---|---|
| | $k=2$ | $k=3$ | $k=2$ | $k=3$ | $k=2$ | $k=3$ | $k=2$ | $k=3$ |
| 1 | 0.0011 | 0.0006 | 0.0011 | 0.0006 | 0.0020 | 0.0010 | 0.0039 | 0.0020 |
| 5 | 0.0030 | 0.0029 | 0.0029 | 0.0030 | 0.0053 | 0.0053 | 0.0099 | 0.0098 |
| 10 | 0.0058 | 0.0059 | 0.0058 | 0.0058 | 0.0104 | 0.0104 | 0.0196 | 0.0196 |
| 25 | 0.0144 | 0.0145 | 0.0145 | 0.0145 | 0.0259 | 0.0260 | 0.0489 | 0.0489 |
| 50 | 0.0289 | 0.0289 | 0.0288 | 0.0289 | 0.0517 | 0.0518 | 0.0975 | 0.0977 |
| 100 | 0.0577 | 0.0577 | 0.0576 | 0.0578 | 0.1033 | 0.1037 | 0.1949 | 0.1953 |

Table 14: Cut value results corresponding to the times of sample $T$ on random regular graphs.

| $T$ | $n=100$ | | $n=1,000$ | | $n=10,000$ | |
|---|---|---|---|---|---|---|
| | $k=2$ | $k=3$ | $k=2$ | $k=3$ | $k=2$ | $k=3$ |
| 0 | 126.71 | 244.77 | 1291.86 | 2408.71 | 12856.53 | 24102.22 |
| 1 | 127 | 245 | 1293 | 2408 | 12856 | 24103 |
| 5 | 127 | 245 | 1293 | 2410 | 12863 | 24103 |
| 10 | 127 | 245 | 1293 | 2410 | 12862 | 24103 |
| 25 | 127 | 245 | 1293 | 2410 | 12864 | 24103 |
| 50 | 127 | 245 | 1293 | 2410 | 12864 | 24103 |
| 100 | 127 | 245 | 1293 | 2410 | 12864 | 24103 |

Table 15: Sampling time results corresponding to the times of sample $T$ on random regular graphs.

| $T$ | $n = 100$ | | $n = 1,000$ | | $n = 10,000$ | |
|---|---|---|---|---|---|---|
| | $k = 2$ | $k = 3$ | $k = 2$ | $k = 3$ | $k = 2$ | $k = 3$ |
| 1 | 0.0001 | 0.0001 | 0.0001 | 0.0001 | 0.0006 | 0.0006 |
| 5 | 0.0006 | 0.0006 | 0.0007 | 0.0007 | 0.0030 | 0.0030 |
| 10 | 0.0011 | 0.0011 | 0.0014 | 0.0013 | 0.0059 | 0.0059 |
| 25 | 0.0026 | 0.0026 | 0.0033 | 0.0031 | 0.0145 | 0.0145 |
| 50 | 0.0052 | 0.0052 | 0.0065 | 0.0060 | 0.0289 | 0.0289 |
| 100 | 0.0103 | 0.0103 | 0.0128 | 0.0122 | 0.0577 | 0.0578 |

