# OpenReview forum: "ROS: A GNN-based Relax-Optimize-and-Sample Framework for Max-$k$-Cut Problems"
_ICML.cc/2025/Conference — ICML 2025 poster_

### Official Review · Reviewer_cFx5 · 2025-03-09

**Overall Recommendation:** 3

**Summary:**

This paper proposes ROS, a GNN-based L2O method, to obtain high-quality max-$k$-cut solutions. The one-hot encoding of each node is relaxed to continuous variables, a GNN is used to do node classification task, i.e., assigning nodes into $k$ partitions, and the continuous output from GNN is then used to construct a feasible solution by a random sampling step. Theoretical result guarantees the existence of feasible max-$k$-cut solution when a globally optimal continuous solution is found. Numerical results on various benchmarks show that ROS can indeed provide high-quality solutions in an efficient way.

## update after rebuttal

My concerns are resolved by those extra experimental results and clarifications. A revised version with those details is acceptable.

**Claims And Evidence:**

Claim 1 [the consistency of function values between continuous solution and its mapped counterpart]: theoretically supported in Theorem 3.2, but lack of empirical evidence. The reason is that Theorem 3.2 requires a global optimum for the relaxation, which is practically learned by a GNN. It will be more convincing if the authors could show the difference between objective values of continuous solutions and their integer counterparts.

Claim 2 [ROS can efficiently scale to large instances]: supported in Section 4.

Claim 3 [ROS exhibits strong generalization capabilities]: supported in Section 4.

**Essential References Not Discussed:**

The discussion about max-$k$-cut is quite sufficient. But the discussion about the idea of relax-optimize-and-sample in other fields is missing.

**Experimental Designs Or Analyses:**

I have several concerns about the experiments.

- The extra cost of preparing training dataset is important to evaluate the effectiveness of a L2O method, which is missing in the paper.

- The experiments only test $k=2,3$, making it unclear if ROS could be applied for larger $k$.

- For the weighted benchmark, the edge weights are constrained to $\pm 1$ with 10\% perturbations. Are there any reasons for choosing such specific settiing? Especially observing that ROS achieves the best performance on this setting compared to other methods.

**Methods And Evaluation Criteria:**

The overall idea of ROS makes sense to me. My only question is about the initial node embeddings. Random embeddings seem to be quite casual and make random seed affect both the training and evaluation. Is it more reasonable to use more meaningful embeddings, e.g., including some neighborhood information?

**Other Comments Or Suggestions:**

- In lines 197-199, after executing Algorithm 1 T times, should one choose the solution with the hightest objective value? if the objective is still defined as Eq. (1).

- The order of references in second column of lines 201-202.

**Other Strengths And Weaknesses:**

**Strengths**

- The paper is well-written and easy to follow.

- Theoretical results are solid.

- Simple setting results in good performance over various scenarios.

**Weaknesses**

I already stated most concerns in above. Additionally, the experimental results are unable to show that ROS could generate better solutions compared to other methods except for faster computational time. The word "high-quality" is quite vague. Intuitively, when the objective value of one solution is approaching to the global optimum, it is much harder to further improve it. It is unclear if ROS already gives meaningful or useful solutions to any practical problems.

**Questions For Authors:**

I already asked all questions in previous sections. They are majorly about clarifications of experiments for better evaluation the importance of ROS, and how well the theoretical contribution is aligned in practice.

**Relation To Broader Scientific Literature:**

ROS follows a standard GNN-based L2O setting. The idea of solving relaxation and then retrieving an integer solution nearby is commonly used in integer programming. The random sampling step as shown in Algorithm 1 is already in the literature, e.g., see https://arxiv.org/pdf/2404.17452.

**Theoretical Claims:**

I roughly checked all proofs and they look correct.

---

> ### Author Rebuttal · Authors · 2025-03-30
>
> ## Claims And Evidence
> **Response to C1:** Theorem 3.2 requires a global optimum, which is why we introduce Theorem 3.3. It theoretically establishes the expected equivalence between relaxed and integer solutions for all feasible points, not just the global optimum. Our sampling algorithm and experiments rely on Theorem 3.3, ensuring practical relevance. Moreover, to stress your concern, we add one row for the continuous objective function in Tables 10 and 12 in the ablation study (updated in Tables 1 and 2 in [anonymous link](https://anonymous.4open.science/r/Tables_For_cFx5-357C/)), explicitly comparing the continuous objcetive values and their integer counterparts.
>
> ## Methods And Evaluation Criteria
> **Response to M1:** See "Response to Q1" for Reviewer zJRf.
>
> ## Experimental Designs Or Analyses
> **Response to E1:** Since ROS is unsupervised, we only generate graphs, not ground truth. As stated in Section 4.1, the training dataset consists of 500 regular graphs, which can be generated within 1 second. The pre-training process runs for only one epoch, requiring just 8.75 seconds in total in our device. In contrast, other L2O baselines, such as ECO-DQN and ANYCSP, demand significantly longer training times, ranging from several hours to multiple days. This highlights the efficiency of ROS in both dataset preparation and model training.
>
> **Response to E2:** We evaluate ROS for larger $k$ (specifically, $k=10$) in our experiments on the real-world Bitcoin-OTC dataset. The corresponding results are provided in Table 1 in "Response to R1" for Reviewer bfPu.
>
> **Response to E3:** We selected the [0.9, 1.1] perturbation range to highlight that ANYCSP struggles even with minimal weight variations, while ROS remains robust across different weight settings in the Max-$k$-Cut problem. To further address your concern, we conduct additional experiments with larger perturbation scales ([0,10] and [0,100]) on the weighted Gset benchmark. The results in Tables 3–6 in [anonymous link](https://anonymous.4open.science/r/Tables_For_cFx5-357C/) show that:
> - ROS consistently achieves the best performance in terms of both solution quality and computational efficiency.
> - Even under extreme perturbations, ROS maintains its advantage over baselines, demonstrating its robustness in handling arbitrary edge weights.
>
> ## Relation To Broader Scientific Literature and Essential References Not Discussed
> While [1] also derives relaxation-based approaches and sampling methods on discrete Bayesian optimization, our sampling step is theoretically derived rather than borrowed directly from prior work. Theorem 3.2 establishes the relationship between continuous and discrete solutions at global optimal, which leads to our sampling strategy designing and analysis: each feasible continuous solution defines a categorical distribution over discrete assignments, and sampling from it preserves the expected objective value (Theorem 3.3). This makes relaxation and sampling inherently connected rather than an arbitrary choice. We acknowledge similar ideas in other fields and will expand the discussion in our paper.
>
> ## Weakness
> As shown in Table 3 of the manuscript and Tables 3–6 in [anonymous link](https://anonymous.4open.science/r/Tables_For_cFx5-357C/), ROS consistently produces the highest-quality solutions while maintaining the fastest computational time in the weighted experiments. Additionally, results on the real-world Bitcoin-OTC dataset (Table 1, response to Reviewer bfPu) demonstrate that ROS effectively handles practical problems, confirming its applicability on weighted Max-$k$-Cut beyond synthetic benchmarks.
>
> ## Other Comments Or Suggestions
> **Response to C1:** We clarify that the correct selection criterion is to choose the solution with the lowest objective value of $f(X)$, as defined in Problem $(P)$ (right column in Line 131), where $f(X)=Tr(XWX^T)$. This corresponds to the highest objective value of the original optimization problem in Equation (1) due to a constant shift and sign inversion. We will revise the manuscript to explicitly state this selection criterion to avoid confusion.
>
> **Response to C2:** We will replace the order of two literatures.
>
> ## Questions for Authors
> - We have added extensive experiments, including tests on real-world weighted Max-$k$-Cut instances and perturbation studies across different ranges. These results, detailed in Table 1 in response to Reviewer bfPu and Tables 3–6 (anonymous link), further highlight the importance of ROS.
> - Regarding the theoretical contribution, ROS is not a direct adaptation from other fields; its components are tightly integrated. The relaxation and sampling steps ensure consistency between the relaxed and discrete solutions, while the powerful GNN effectively bridges the optimization gap. This demonstrates a strong alignment between theory and practical performance.
>
> ## Reference
> [1] Michael R, et al. A Continuous Relaxation for Discrete Bayesian Optimization[J].

---

> > ### Comment · Reviewer_cFx5 · 2025-04-04
> >
> > Thanks for your response. Those extra experimental results and clarifications resolve most of my concerns. I will raise my rating to 3 and suggest the authors add those results properly in a revised version.

---

> > > ### Author Response · Authors · 2025-04-04
> > >
> > > > **Comment:** Thanks for your response. Those extra experimental results and clarifications resolve most of my concerns. I will raise my rating to 3 and suggest the authors add those results properly in a revised version.
> > >
> > > **Reply:** We appreciate your efforts in reviewing our paper and rebuttal. Thank you for your feedback and for raising the rating. We will incorporate the additional experimental results and clarifications into the revised version to further strengthen the paper. Thank you again for your constructive comments!

---

### Official Review · Reviewer_rFKE · 2025-03-11

**Overall Recommendation:** 3

**Summary:**

The paper introduces, a GNN-based framework for solving the Max-k-Cut problem by relaxing the discrete optimization problem into a continuous optimization task. A Graph Neural Network (GNN) optimizes the relaxed problem, followed by a sampling-based algorithm to obtain a discrete solution. The authors integrate geometric landscape analysis with statistical theory to establish the consistency of function values between the continuous solution and its mapped discrete counterpart.

**Claims And Evidence:**

Authors show the superiority of their algorithms but it is not clear if baselines involving learning base techniques also used the pretraining and finetuning phase or not.

In addition, some direct baselines have neither been cited nor compared with.

**Essential References Not Discussed:**

As mentioned above, important related works have not been disucussed and compared to.

[1] Rishi Rajesh Shah, Krishnanshu Jain, Sahil Manchanda, Sourav Medya and Sayan Ranu, "NeuroCut: A Neural Approach for Robust Graph Partitioning", in KDD, 2024.

[2] Anton Tsitsulin, John Palowitch, Bryan Perozzi, and Emmanuel Müller. 2023. Graph clustering with graph neural networks. Journal of Machine Learning Research 24, 127 (2023), 1–21.

[3] Aritra Bhowmick, Mert Kosan, Zexi Huang, Ambuj Singh, and Sourav Medya. 2024. DGCLUSTER: A Neural Framework for Attributed Graph Clustering via Modularity Maximization. In Proceedings of the AAAI Conference on Artificial Intelligence, Vol. 38. 11069–11077.

**Experimental Designs Or Analyses:**

- Could you please clarify whether Fig. 2 represents the training time or inference time. Could you please demonstrate scalability with respect to ground-truth generation, training time and inference time explicitly and compare with non-neural approaches. Non-neural approaches generalize to any value of $k$. Hence, it is important to look at the scalability of all three dimensions to evaluate the practical value of this work.

**Methods And Evaluation Criteria:**

- **Baselines:** It appears [1][2] and [3] are related and potential baselines, by suitably changing the loss/reward function. [1], in particular, does not even need the optimization function to be differentiable. Why are they not discussed and compared with?

[1] Rishi Rajesh Shah, Krishnanshu Jain, Sahil Manchanda, Sourav Medya and Sayan Ranu, "NeuroCut: A Neural Approach for Robust Graph Partitioning", in KDD, 2024.

[2] Anton Tsitsulin, John Palowitch, Bryan Perozzi, and Emmanuel Müller. 2023. Graph clustering with graph neural networks. Journal of Machine Learning Research 24, 127 (2023), 1–21.

[3] Aritra Bhowmick, Mert Kosan, Zexi Huang, Ambuj Singh, and Sourav Medya. 2024. DGCLUSTER: A Neural Framework for Attributed Graph Clustering via Modularity Maximization. In Proceedings of the AAAI Conference on Artificial Intelligence, Vol. 38. 11069–11077.

- **Datasets:** None of the datasets are real. Please include real world datasets containing atleast few thousands of nodes (See [1] for example)

- **Generalizability to $k$:** It appears that the method needs to know the number of partitions ($k$) before inference. Specifically, it needs to train for each specific value of $k$ since it does not generalize to unseen $k$ at inference time. This is evident from line 232 where the output embedding is $\mathbb{R}^{k\times N}$.

**Other Comments Or Suggestions:**

In Tables 2,3, and 4, you can directly put ROS without finetuning results. It will enhance readability.

**Other Strengths And Weaknesses:**

**Strengths:**

- The paper presents a GNN-based framework for solving the weighted Max-k-Cut problem, converting a discrete optimization task into a continuous one for easier processing, which is interesting and novel

- Compared to the baselines, the results look good both in efficiency and quality.

**Weakness:**

- The authors have not mentioned if leaning-based baselines are used pertaining and fine-tuning steps separately or not.
- Figure 2 appears to be inference time. Please report time for pre-training, training/fine-tuning, etc.
- The method does not generalize to unseen $k$. This appears to be a serious limitation.
- The benchmark datasets do not include any real-world dataset
- Important works have not been discussed and compared to.
- Code base is not shared and hence reproducibility is hampered.

**Questions For Authors:**

The key reasons for my current rating are below. I would be happy to revisit the rating if the questions raised below are satisfactorily addressed.

1Justify why the inability to generalize to unseen $k$ during inference is not a severe limitation.
2. Please discuss (and compare with unless there are obvious reasons not to) the missing baselines discussed above.
3. Include real world datasets.

**Relation To Broader Scientific Literature:**

Earlier works try to solve maximum-k-cut problems using graph learning approaches but they seem limited to the unweighted setting while the proposed approach deals with solving weighted maximum-k-cut problems using GNN.

**Theoretical Claims:**

Seems intuitively correct but did not go through deeply.

---

> ### Author Rebuttal · Authors · 2025-03-30
>
> ## Claims And Evidence
> **Response to C1:** Please refer to "Response to W2" for Reviewer bfPu.
>
> ## Methods And Evaluation Criteria
> **Response to M1:**
> - NeuroCUT [1] is a reinforcement learning-based partitioning method, while DGCLUSTER [2] and DMoN [3] employ graph neural networks to optimize clustering objectives. However, these methods are designed for graph clustering, which aims to minimize inter-cluster connections, whereas Max-$k$-Cut seeks to maximize inter-partition connections. As a result, they are not directly applicable to our problem. Additionally, while NeuroCUT claims to support arbitrary objective functions, its node selection heuristics are only tailored for graph clustering, **making it unsuitable for Max-$k$-Cut**.
> - Despite these differences, we evaluated NeuroCUT as a representative baseline of graph clustering. We trained it on 500 3-regular graphs as ROS and tested it on Bitcoin-OTC, a real-world signed network with 5,881 nodes and 35,592 weighted edges (ranging from -10 to 10), which captures trust relationships among Bitcoin traders. **The results is shown in "Response to R1" for Reviewer bfPu**, ROS significantly outperforms NeuroCUT and other baselines, further demonstrating its effectiveness for Max-$k$-Cut.
>
> **Response to M2:** We include a real-world dataset, Bitcoin-OTC, in our evaluation, which contains 5,881 nodes and 35,592 weighted edges. The comparison results with baselines on this dataset are presented in "Response to R1" for Reviewer bfPu.
>
> **Response to M3:** Please refer to the "response to W1" of reviewer bfPu for details regarding the generalizability of our method to unseen $k$.
>
> ## Experimental Designs Or Analysis:
> Figure 2 represents the fine-tuning time for ROS. ROS does not require ground-truth generation, unlike supervised methods. Pre-training for a specific $k$ is lightweight—training on 500 regular graphs for one epoch takes only 8.75 seconds, whereas L2O baselines like ECO-DQN and ANYCSP require hours or even days. The scalability of fine-tuning (inference) time is detailed in Table 1 of the manuscript, and ROS efficiently scales to instances of large $N$.
>
>
>
> ## Essential References Not Discussed
> Please see the response to M1.
>
> ## Weakness
> **Response to W1:** Please see "Response to W2" for Reviewer bfPu.
>
> **Response to W2:** As stated in Section 4.1, the training dataset consists of 500 regular graphs, and the pre-training process runs for only one epoch, requiring just 8.75 seconds in total. In contrast, L2O baselines like ECO-DQN and ANYCSP require significantly longer training times, ranging from several hours to multiple days. This highlights the efficiency of ROS. The fine-tuning (inference) time is already reported in Section 4.
>
> **Response to W3:** Please refer to the "response to W1" for reviewer bfPu regarding the generalizability of our method to unseen $k$.
>
> **Response to W4:** Please see Response to M2.
>
> **Response to W5:** Please see Response to M1.
>
> **Response to W6:** We upload our code in https://anonymous.4open.science/r/ROS_anonymous-1C88/.
>
>
> ## Other Comments Or Suggestions
> Since fine-tuning directly solves test instances, we cannot remove this stage. However, to enhance readability, we now include results for ROS-vanilla (i.e., ROS without pre-training). The updated tables explicitly present ROS-vanilla results, improving clarity. Below are the updated rows in Tables 1, 2, and 3 of the manuscript (Tables 4 and 5 already included ROS-vanilla results):
>
> **Updated Row in Table 1 in manuscripts**
> | Model| $N=100, k=2$| $N=100, k=3$| $N=1000, k=2$| $N=1000, k=3$| $N=10000, k=2$| $N=10000, k=3$      |
> | - | - | - | - | - | - | - |
> | ROS-vanilla | $132.00\pm 1.89$ | $243.75\pm 2.00$ | $1322.95\pm 6.57$ | $2440.55\pm 4.97$ | $13191.25\pm 20.73$ | $24317.40\pm 21.36$ |
>
> **Updated Row in Table 2 in manuscripts**
> | Model| G70 ($k=2$) | G70 ($k=3$) | G72 ($k=2$) | G72 ($k=3$) | G77 ($k=2$) | G77 ($k=3$) | G81 ($k=2$) | G81 ($k=3$) |
> | - | - | - | - | - | - | - | - | - |
> | ROS-vanilla | 9004| 9982| 6066| 7210| 8678        | 10191| 12260| 14418|
>
> **Updated Row in Table 3 in manuscripts**
> | Model| G70 ($k=2$) | G70 ($k=3$) | G72 ($k=2$) | G72 ($k=3$) | G77 ($k=2$) | G77 ($k=3$) | G81 ($k=2$) | G81 ($k=3$) |
> | - | - | - | - | - | - | - | - | - |
> | ROS-vanilla | 8989.38| 9973.75| 6140.50| 7207.13| 8744.47| 10190.37| 12278.70| 14341.25|
>
> ## Questions For Authors
> **Response to Q1:** Please refer to the "response to W1" for reviewer bfPu.
>
> **Response to Q2:** Please see Response to M1.
>
> **Response to Q3:** Please see Response to M2.
>
> ## Reference
> [1] Rishi Rajesh Shah et al., NeuroCut: A Neural Approach for Robust Graph Partitioning, in KDD.
> [2] Anton Tsitsulin et al., Graph clustering with graph neural networks. In JMLR.
> [3] Aritra Bhowmick et al, DGCLUSTER: A Neural Framework for Attributed Graph Clustering via Modularity Maximization. In AAAI.

---

> > ### Comment · Reviewer_rFKE · 2025-04-02
> >
> > The generalization to unseen $k$ seems like a hack. I am happy with the other changes made and will increase the rating to 3.

---

> > > ### Author Response · Authors · 2025-04-03
> > >
> > > > **Comment:** The generalization to unseen $k$ seems like a hack. I am happy with the other changes made and will increase the rating to 3.
> > >
> > > **Response:** We appreciate your efforts in reviewing our paper and rebuttal. We also thank you for your feedback and consideration in raising the rating. Regarding the generalization to unseen $k$, we provide two approaches based on the "pre-training + fine-tuning" framework of ROS:
> > >
> > > - **ROS-vanilla**: This method is directly fine-tuned on the test instance without pre-training, avoiding dependency on predefined last-layer dimensions.
> > >
> > > - **ROS-partial**: To apply the pre-training technique and improve efficiency while still generalizing to unseen $k$, this variant is pre-trained on $k=2$ while saving all parameters except the last layer. Before fine-tuning, the pre-trained parameters are loaded, and the last layer is randomly initialized to accommodate the new $k$.
> > >
> > > As shown in Table 2 in response to Reviewer bfPu, both approaches demonstrate the **flexibility and extensibility** of our "pre-train + fine-tune" framework of ROS. Furthermore, while our framework supports generalization to unseen $k$, we acknowledge that exploring this aspect through model architecture design, as in [1], is an exciting direction. We appreciate your valuable comments once again.
> > >
> > > [1] NeuroCUT: A Neural Approach for Robust Graph Partitioning Rishi Shah, Krishnanshu Jain, Sahil Manchanda, Sourav Medya, Sayan Ranu KDD Knowledge Discovery and Data Mining(KDD), 2024.

---

### Official Review · Reviewer_bfPu · 2025-03-15

**Overall Recommendation:** 4

**Summary:**

This paper introduces ROS, a GNN-based framework for Max-k-Cut. The authors propose a solution that relaxes the problem to a continuous space, optimizes it with a neural network, and samples a discrete solution. They compare with existing neural and non neural baselines and show they are better in terms of quality and running time.

**Claims And Evidence:**

1. ROS has better quality . Supported by evaluation on diverse dataset and values of k.
2. Better running time. supported by running time plots.

**Essential References Not Discussed:**

[A] NeuroCUT: A Neural Approach for Robust Graph Partitioning
Rishi Shah, Krishnanshu Jain, Sahil Manchanda, Sourav Medya, Sayan Ranu
KDD Knowledge Discovery and Data Mining(KDD), 2024

[A] solves graph partitoning problem for arbitrary partitoning objectives. The approach is inductive to number of partitioing.

**Experimental Designs Or Analyses:**

Mostly it is clear. Only thing that is not clear whether baselines were fine-tuned?

**Methods And Evaluation Criteria:**

Yes

**Other Comments Or Suggestions:**

None

**Other Strengths And Weaknesses:**

Weakness:
1. Kindly clarify if model can do inference on unseen k(number of partitions). Can the model be fine-tuned to different k? If yes, how?
From line 233 it seems output layer is fixed to k.

2. Were the neural baselines also fine-tuned? Kindly clarify.

3. Code is not shared.

**Questions For Authors:**

Check weakness

**Relation To Broader Scientific Literature:**

1. Proposed method does not require any ground truth.
2. Framework uses relaxation approach, which is effective in this setup.

**Theoretical Claims:**

Did not check proofs in details.

---

> ### Author Rebuttal · Authors · 2025-03-30
>
> ## Essential Reference Not Discussed
>
> **Response to R1:**
> - NeuroCUT [1] is a reinforcement learning-based partitioning method designed for graph clustering, which aims to minimize inter-cluster connections, whereas Max-$k$-Cut seeks to maximize inter-partition connections. Additionally, while NeuroCUT claims to support arbitrary objective functions, its node selection heuristics are specifically tailored for graph clustering. Due to this fundamental difference, NeuroCUT is not directly applicable to our problem.
> - Despite these differences, we compared our methods with NeuroCUT. We trained NeuroCUT on 500 3-regular graphs as ROS and tested it on Bitcoin-OTC [2], a real-world signed network with 5,881 nodes and 35,592 weighted edges (ranging from -10 to 10), which captures trust relationships among Bitcoin traders. As shown in Table 1, ROS significantly outperforms NeuroCUT and other baselines, further demonstrating its effectiveness for Max-$k$-Cut.
>
> **Table 1: Evaluation results on Bitcoin-OTC Datasets.**
> | Model    | Value ($k=2$) | Time (s) ($k=2$) | Value ($k=3$) | Time (s) ($k=3$) | Value ($k=10$) | Time (s) ($k=10$) |
> | - | - | - | - | - | - | - |
> | NeuroCut |1424|239.46| 1667| 242.65| 13235| 250.90|
> | PIGNN| 14587| 62.31| -| -| -| -|
> | MD| 14989| 37.15| 18448| 50.40| 21182| 105.92|
> | ANYCSP| 10678| 180.20| 14319| 180.16| 19359|180.24|
> | ROS| **15384**| **2.94**| **18585**| **2.44**| **21251**| **2.04**|
>
> ## Weakness
>
> **Response to W1:**
>
> - **ROS-vanilla** (without pre-training) can directly generalize to any value of $k$ since the absence of pre-training.
>
> - **ROS** (with pre-training and fine-tuning) improves efficiency but does not generalize directly to unseen $k$ due to the fixed output layer during pre-training. However, this limitation can be addressed through **ROS-partial**, a simple modification that enables adaptation to different $k$.
>
> - **ROS-partial** works by pre-training the model on $k=2$ while saving all parameters except the last layer. Before fine-tuning, the pre-trained parameters are loaded, and the last layer is randomly initialized to accommodate the new $k$. This approach serves as a middle ground between ROS (fully pre-trained) and ROS-Vanilla (no pre-training).
>
> - We evaluate **ROS-partial**, **ROS**, and **ROS-vanilla** on the Bitcoin-OTC dataset. The results in Table 2 show that ROS-partial effectively generalizes to different $k$ while maintaining strong performance.
>
> **Table 2: Comparison between pre-training ways on Bitcoin-OTC Datasets.**
> | Model    | Value ($k=2$) | Time (s) ($k=2$) | Value ($k=3$) | Time (s) ($k=3$) | Value ($k=10$) | Time (s) ($k=10$) |
> | - | - | - | - | - | - | - |
> | ROS| 15384| **2.94**| 18585| **2.44**| 21251| **2.04**|
> | ROS-vanilla| **15661**| 5.24| **18977**| 4.77| **21365**| 4.43|
> | ROS-partial| 15102| 4.24| 18732| 3.93| 21308| 2.92|
>
>
>
> **Response to W2:**
> - The pre-training and fine-tuning phases of ROS correspond to the training and inference phases of other L2O baselines. Specifically, ROS is pre-trained on a collected dataset, similar to how L2O baselines are trained. During fine-tuning, ROS further optimizes based on **test instances**, whereas standard L2O inference keeps parameters fixed. To ensure fairness, we include the full fine-tuning time in our reported results. Thus, the datasets used for pre-training (training) and fine-tuning (testing) in ROS align with those in other L2O methods.
> - To further address the concern, we also conduct experiments where we introduce fine-tuning to existing L2O baselines. After training, these models are further fine-tuned on test instances, and we plot the cut value against fine-tuning iterations in the [anonymous link](https://anonymous.4open.science/r/Figures_for_bfPu-9473). The results confirm that even with fine-tuning, other baselines do not surpass ROS in solution quality.
>
> **Response to W3:** We upload our code in https://anonymous.4open.science/r/ROS_anonymous-1C88/.
>
> ## Reference
> [1] Rishi Rajesh Shah, Krishnanshu Jain, Sahil Manchanda, Sourav Medya and Sayan Ranu, "NeuroCut: A Neural Approach for Robust Graph Partitioning", in KDD, 2024.
>
> [2] S. Kumar, F. Spezzano, V.S. Subrahmanian, C. Faloutsos. Edge Weight Prediction in Weighted Signed Networks. IEEE International Conference on Data Mining (ICDM), 2016.

---

> > ### Comment · Reviewer_bfPu · 2025-04-04
> >
> > I thank the authors for additional experiments.
> > Could you clarify what was the objective used for NeuroCUT in this experiment.
> > Further, the node selection heuristic I believe could be kept random in this case.
> >
> > I would expect clarification on how was NeuroCUT was integrated just to ensure comparison is fair.
> >
> > I am happy to see running time results. ROS is significantly faster than all methods.
> > Although generalization to k is hacky, but happy to see the results.

---

> > > ### Author Response · Authors · 2025-04-05
> > >
> > > > **Comment:** I thank the authors for additional experiments. Could you clarify what was the objective used for NeuroCUT in this experiment. Further, the node selection heuristic I believe could be kept random in this case. I would expect clarification on how was NeuroCUT was integrated just to ensure comparison is fair. I am happy to see running time results. ROS is significantly faster than all methods. Although generalization to k is hacky, but happy to see the results.
> > >
> > > **Reply:** We appreciate your efforts in reviewing our paper and rebuttal, and thank you for your valuable feedback.
> > > - To implement NeuroCUT, we used the loss function defined in problem $(P)$ (line 131, right column), as the source code of NeuroCUT minimizes the objective.
> > > - To ensure a fair comparison, we replaced the original score-based node selection heuristic with random selection, as suggested.
> > > - Additionally, we found that the original K-means initialization is designed for graph clustering, and applying it to Max-$k$-Cut often leads to suboptimal starting points, even worse than random initialization. Therefore, we replaced it with random initialization.
> > >
> > > The updated results on Bitcoin-OTC are included in the following table.
> > >
> > > **Updated Table 1: Updated Evaluation results on Bitcoin-OTC Datasets. Here, NeuroCut is equipped with random initialization and random node selection, which is different from the previous Table 1.**
> > > | Model    | Value ($k=2$) | Time (s) ($k=2$) | Value ($k=3$) | Time (s) ($k=3$) | Value ($k=10$) | Time (s) ($k=10$) |
> > > | - | - | - | - | - | - | - |
> > > | NeuroCut| 10260 | 240.98 | 10896 | 237.09 | 17768 | 249.99 |
> > > | PIGNN| 14587| 62.31| -| -| -| -|
> > > | MD| 14989| 37.15| 18448| 50.40| 21182| 105.92|
> > > | ANYCSP| 10678| 180.20| 14319| 180.16| 19359|180.24|
> > > | ROS| **15384**| **2.94**| **18585**| **2.44**| **21251**| **2.04**|
> > >
> > > Regarding the generalization to unseen $k$, we provide two approaches based on the "pre-training + fine-tuning" framework of ROS:
> > >
> > > - **ROS-vanilla**: This method is directly fine-tuned on the test instance without pre-training, avoiding dependency on predefined last-layer dimensions.
> > >
> > > - **ROS-partial**: To apply the pre-training technique and improve efficiency while still generalizing to unseen $k$, this variant is pre-trained on $k=2$ while saving all parameters except the last layer. Before fine-tuning, the pre-trained parameters are loaded, and the last layer is randomly initialized to accommodate the new $k$.
> > >
> > > As shown in Table 2 of the rebuttal, both approaches demonstrate the **flexibility and extensibility** of our "pre-train + fine-tune" framework of ROS. Furthermore, while our framework supports generalization to unseen $k$, we acknowledge that exploring this aspect through model architecture design, as in [1], is an exciting direction.
> > >
> > > We appreciate your valuable comments once again.
> > >
> > > [1] NeuroCUT: A Neural Approach for Robust Graph Partitioning Rishi Shah, Krishnanshu Jain, Sahil Manchanda, Sourav Medya, Sayan Ranu KDD Knowledge Discovery and Data Mining(KDD), 2024.

---

### Official Review · Reviewer_zJRf · 2025-03-21

**Overall Recommendation:** 3

**Summary:**

The paper proposes a GNN-based solver for the mak-k-cut problem.

**Claims And Evidence:**

Yes. But I do have many questions.

**Essential References Not Discussed:**

n/a

**Experimental Designs Or Analyses:**

See comments.

**Methods And Evaluation Criteria:**

Some points may not be clear enough. For example,

- Does other baselines use the same training data as the training+finetuning datasets of ROS? If not, can other methods be fit under the pretrain-finetune framework?

**Other Comments Or Suggestions:**

- Fig. 1 has two h6 in the initialization step of the grey box

- Notations are sometimes abused. For example, \overline{X} shows up three times with different meanings. In Def 3.1, it is a point. In Theorem 3.2, it is the globally optimal solution. In Q2, it is a high quality solution. They make me confused.

- Statistics of the datasets are not given. For example, how many graphs are in the different datasets?

- In the results of figure 2, which part of the datasets are used for training, which part are for finetuning, and which part are for testing? It seems that Sec. 4.1 does not figure it out.

**Other Strengths And Weaknesses:**

Strengths

- The max k cut problem, as a generalization of the max cut problem, holds significant research value.

Weaknesses

- The scope of the paper is relatively narrow. The proposed framework is only applicable to the max k cut problem. It would be more meaningful if the framework could be more widely applied to other combinatorial optimization problems.

- The method appears to lack novelty. The pretrain-finetune framework employed is not new and has been widely used in other contexts.

- The comparison with baseline methods may not be entirely fair. I question whether the training data used are consistent with those used in ROS. Given that ROS involves a two-stage process of pretraining and finetuning, whereas other methods only include a single training step, this discrepancy could affect the validity of the comparisons.

- I find the experimental section somewhat challenging to follow. The descriptions of the experimental settings are somewhat disorganized and could benefit from clearer and more structured presentation.

- The absence of a related works section is notable. While it is true that research on the max k cut problem may be relatively sparse, it is still unusual to omit this section entirely. Including a discussion of related works would provide valuable context and help situate the current research within the broader field.

**Questions For Authors:**

- Noticing that the initial embeddings h0 are assigned by random values. I have doubt on the correctness of doing so.

**Relation To Broader Scientific Literature:**

The work contributes to the combinatorial optimization community.

**Theoretical Claims:**

I did not check them very carefully.

---

> ### Author Rebuttal · Authors · 2025-03-30
>
> ## Methods And Evaluation Criteria
> **Response to M1:** Please see "Response to W2" for Reviewer bfPu.
>
> ## Weakness
> **Response to W1:** The Max-$k$-Cut problem is a fundamental NP-complete problem with applications in physics [1], power networks [2], and data clustering [3]. While ROS is tailored for Max-$k$-Cut, its core Relax-Optimize-and-Sample framework is generalizable to other combinatorial optimization problems by adjusting the objective functions. Investigating such extensions, particularly their theoretical guarantees, is a promising direction for future work.
>
> **Response to W2:** The core novelty of our work lies in the Relax-Optimize-and-Sample (ROS) framework, where the "pre-train + fine-tune" approach is used solely for efficiency. The key contributions of ROS are:
> - The probability simplex relaxation ensures that the optimal values of the relaxed and original Max-$k$-Cut problems are equivalent (Theorem 3.2).
> - A GNN parametrizes the decision variable, enhancing both representation power and computational efficiency.
> - The proposed sampling procedure maps the relaxed solution to a discrete Max-$k$-Cut solution while preserving the objective value (Theorem 3.3).
>
> **Response to W3:** Please see "Response to W2" for Reviewer bfPu.
>
> **Response to W4:** We will refine and reorganize the experimental section to improve clarity and readability. Specifically, we will integrate Tables 2 and 3 in [anonymous link](https://anonymous.4open.science/r/Tables_and_Reference_For_zJRF-9D0B) into the manuscript to provide a clearer presentation of dataset statistics and enhance the overall structure of the experimental setup.
>
> **Response to W5:** We will add a Related Work section to provide context. This will cover approximation algorithms (e.g., Goemans-Williamson (GW) [4], Frieze et al. [5]), non-convex relaxations (Rank-2 [6], QUBO [7]), and Lovász extensions [8]. We will clarify how ROS differs by ensuring objective value consistency and leveraging GNN-based optimization for high-quality solutions. We will also add Table 1 in [anonymous link](https://anonymous.4open.science/r/Tables_and_Reference_For_zJRF-9D0B) to further clarify the distinctions between these methods.
>
> ## Other Comments and Suggestions:
> **Response to C1 and C2:** We will revise Fig. 1 to correct the typo and ensure that all notations are used consistently throughout the manuscript.
>
> **Response to C3:** The statistics of the training and testing datasets are summarized in Tables 2 and 3 in [anonymous link](https://anonymous.4open.science/r/Tables_and_Reference_For_zJRF-9D0B), which we will add to the manuscript for clarity.
>
> **Response to C4:** The training dataset consists of 500 3-regular graphs for $k=2$ and 500 5-regular graphs for $k=3$. The fine-tuning (testing) datasets correspond to different graph types: (a) random regular graphs, (b) Gset, and \(c) weighted Gset, as detailed in Section 4.1 (Line 272, right column, Page 5).
>
> ## Questions For Authors
> **Response to Q1:**
> - The random initialization does not introduce instability, as shown by the low standard deviation in Table 1 in manuscripts, where the relative error across runs remains around 1%.
> - Additionally, the node features and the adjacency information can be incorporated as initialization in ROS when it is available. For example, on the Cora dataset [9], we interpolated both node features as well as the adjacency matrix to match the input dimension. Results in Table 4 in the [anonymous link](https://anonymous.4open.science/r/Tables_and_Reference_For_zJRF-9D0B) show that when all other model parameters remain the same, the model yields identical outputs across different initialization methods, confirming the adequacy of random initialization.
> - Furthermore, random initialization facilitates distributed deployment where each node is employed on a different device, avoiding global operations required for feature interpolation.
>
> ## Reference
>
> See [anonymous link](https://anonymous.4open.science/r/Tables_and_Reference_For_zJRF-9D0B).

---

### Decision · Program_Chairs · 2025-05-01

**Decision:**

Accept (poster)

**Comment:**

This submission proposes the Relax-Optimize-and-Sample (ROS) framework to solve the Max-k-Cut problem. Unlike relaxation techniques that do not guarantee equivalence between relaxation solutions and solutions of the original problem, ROS involves a sampling-based construction algorithm to map the relaxed solution (produced using a GNN) back to a high-quality solution in the original problem space. The authors provide theoretical guarantees on consistency. Experiments in settings of up to 20K nodes show fast computational time, and reasonable performance (while ROS does not "dominate" with respect to every experiment, its benefits are demonstrated with respect to generalizability across instances).

Reviewers initially raised concerns about the experimental setups (lack of real-world experiments) and fairness of baseline comparisons, but these concerns were mitigated during author-reviewer discussion, notably via the addition of a real-world Bitcoin-OTC dataset (with $k=10$). They also expressed some concerns about the clarity of the experiments section (which should be addressed in the revised version). However, overall, reviewers were convinced of the relevance and significance of the contribution.